# Molecular Epidemiology and In-Depth Characterization of *Klebsiella pneumoniae* Clinical Isolates from Armenia

**DOI:** 10.3390/ijms26020504

**Published:** 2025-01-09

**Authors:** Anahit Sedrakyan, Zaruhi Gevorgyan, Magdalina Zakharyan, Karine Arakelova, Shoghik Hakobyan, Alvard Hovhannisyan, Rustam Aminov

**Affiliations:** 1Institute of Molecular Biology, National Academy of Sciences of RA, Yerevan 0014, Armenia; sedanahit@gmail.com (A.S.); linazakharyan@gmail.com (M.Z.); karinaraqel@gmail.com (K.A.); shoghik.hakobyan.96@gmail.com (S.H.); alla_hov@yahoo.com (A.H.); 2Department of Clinical Laboratory Diagnostics, Yerevan State Medical University After M. Heratsi, Yerevan 0025, Armenia; zaragevorgyan@yahoo.de; 3School of Medicine, Medical Sciences and Nutrition, University of Aberdeen, Aberdeen AB25 2ZD, UK

**Keywords:** *Klebsiella pneumoniae*, antimicrobial resistance, carbapenem resistance, ERIC-PCR typing, WGS, resistome, virulome, mobile genetic elements

## Abstract

The global dissemination of *Klebsiella pneumoniae* pathotypes with multidrug-resistant (MDR) and hypervirulent traits poses a threat to public health. The situation in Armenia is unclear, and we performed a comprehensive characterisation of 48 clinical isolates of *K. pneumoniae*, collected from 2018 to 2024. The majority of the isolates (64.58%) were extensively drug-resistant (XDR) and MDR. Genomic analysis of 21 isolates revealed the presence of international high-risk MDR clones (ST395, ST15, and ST307). The ST395 strains were isolated from children and resisted the first-line drugs such as beta-lactams. These isolates harboured a range of virulence determinants, from capsule polysaccharides to siderophores to regulators of the mucoid phenotype. The ST395 strains are enriched by ICEs, plasmids, and prophages, on which antimicrobial resistance (AMR) and virulence genes are located and which may lead to the convergence of MDR and hypervirulent traits. There is a widespread non-specific AMR mechanism among our *K. pneumoniae* strains. These are mutations in the porin genes, which reduce permeability to antimicrobials, and mutations in the regulators of efflux pumps, which lead to overexpression of drug efflux pumps such as AcrAB. These mechanisms may contribute to the elevated MICs and confer AMR to strains with no specific AMR genes.

## 1. Introduction

Emergence and rise of antimicrobial resistance (AMR) in pathogenic bacteria have become a significant public health threat over the last several decades, and this problem continues to escalate. The first comprehensive assessment of the global burden of AMR highlighted the excessive mortality rates associated with AMR infections [1]. In a recent response to this alarming trend, the World Health Organisation (WHO) has published an updated list of priority AMR pathogens, for which new therapeutic alternatives are urgently needed [2]. Among other priority pathogens, the list includes carbapenem-resistant and cephalosporin-resistant Enterobacterales. The increasing resistance to carbapenems and cephalosporins in these bacteria compromises the efficacy of therapy and limits treatment options for patients facing challenging infections.

Among Enterobacterales, *Klebsiella pneumoniae* has been estimated to be second only to *Escherichia coli* in the mortality rates directly attributable to AMR, with 193 and 219 million deaths globally, respectively [1]. *K. pneumoniae* is considered one of the most widespread opportunistic bacteria that cause pneumonia, urinary tract infections, meningitis, sepsis, and other life-threatening diseases [3,4]. Its pathogenicity mechanisms, however, are still poorly understood. Another problem with this bacterium is its multidrug resistance (MDR). It belongs to the ESKAPE bacteria (*Enterococcus faecium*, *Staphylococcus aureus*, *K. pneumoniae*, *Acinetobacter baumannii*, *Pseudomonas aeruginosa*, and *Enterobacter* species), for which the therapeutic options are limited because of the widespread MDR phenotypes [5]. Thus, the attention to MDR *K. pneumoniae* is eminent because of its clinical significance as one of the leading pathogens associated with nosocomial infections that are particularly difficult to treat, especially in patients who are in the risk groups such as elderly, neonates, and patients with chronic diseases or immunocompromised [6,7,8,9,10]. Another important pathotype includes hypervirulent *K. pneumoniae* (hvKp) strains, which are susceptible to antimicrobials but can affect healthy individuals and may cause severe community-acquired infections such as pyogenic liver abscess with metastatic infections, pneumonia, urinary tract infections, and other diseases [3,6,10,11,12,13]. The prevalence of this pathotype remains underestimated since the reliable biomarkers for hvKp are still in development [14,15], and detection of hypervirulence is not commonly performed in a standard clinical microbiology laboratory. Furthermore, the emergence of convergent *K. pneumoniae* strains that combine MDR and hvKp phenotypes has been increasingly reported [10,16,17]. Dissemination of this emerging pathotype may pose a global public health threat [18,19]. Of special concern are the *K. pneumoniae* lineages exhibiting both hvKp and carbapenem resistance phenotypes [18,19,20]. This drastically limits treatment options and necessitates the use of the last-resort antimicrobials (AMs) with known side effects such as colistin because the alternative antimicrobial therapies [21] are not always available.

The global dissemination of epidemically important high-risk clones of *K. pneumoniae* poses a significant threat that requires extensive monitoring and appropriate control measures. Especially important is the genomic surveillance since there is growing evidence indicating a remarkable genetic diversity among *K. pneumoniae* strains. This diversity is mainly driven by horizontal gene exchange, and it results in the emergence of a variety of clones at different times and in different regions [9,19]. This clonal diversity changes the landscape of difficult-to-treat or/and hvKp infections and challenges effective infection prevention and control. To deal with this threat, many countries implemented monitoring programs, which help to understand the epidemiology and drug resistance of this pathogen and take the necessary measures to treat and control it. In particular, this information is especially useful for health professionals, who deal with severely infected patients and have to make a prompt decision regarding empirical AM therapy, which should be the most appropriate under the current regional circumstances. In some countries and regions, however, there is a paucity of information regarding the genomic structure of local *K. pneumoniae* pathotypes. In particular, it is evident for Armenia, where this information is very scarce [22,23,24]. There is only a single published report concerning the genomic study of eight MDR *K. pneumoniae* clinical isolates from Armenia [24]. The aim of this study, therefore, was to explore the molecular epidemiology, resistome, and virulome of *K. pneumoniae* clinical isolates in Armenia. We attempted to characterise these *K. pneumoniae* isolates in depth, including various phenotypic and genomic characteristics, to present the combined data for appropriate public health measures at the local and regional levels.

## 2. Results

### 2.1. Antimicrobial Susceptibility Among Clinical Isolates of K. pneumoniae

Among our 48 clinical isolates of *K. pneumoniae*, the highest rates of susceptibility were detected towards colistin (93.75%, 45/48, MIC ≤ 1 µg/mL) and tigecycline (93.75%, 45/48, MIC ≤ 0.5 µg/mL), followed by meropenem (89.58%, 43/48) and imipenem (89.58%, 43/48) (Figure 1).

Among cephems, the most effective was cefoxitin, with a 79.17% (38/48) susceptibility rate, while 6.25% (3/48) and 14.58% (7/48) of isolates displayed intermediate and full resistance, correspondingly (Figure 1). Susceptibility to other AMs of this class was 56.25% (27/48) for cefepime, 50% (24/48) for ceftazidime, and 47.92% (23/48) for ceftriaxone. These findings indicate the high prevalence of resistance to the 3rd and 4th generation cephalosporins among our isolates. The similar levels of susceptibility were found towards all beta-lactam combination agents tested in our study; the most efficient was ticarcillin-clavulanate, with a 50% (24/48) susceptibility rate. Susceptibility to another beta-lactam, aztreonam, was also in the same range (52.08%, 25/48).

As for azithromycin, 66.67% (32/48) of isolates had minimum inhibitory concentrations (MICs) of ≤16 µg/mL, whereas the remaining 33.33% (16/48) of isolates displayed MICs higher than 64 µg/mL (Figure 1). The rates of susceptibility to other AMs, in descending order, were as follows: 77.08% (37/48) for amikacin, 64.58% (31/48) for chloramphenicol, 56.25% (27/48) for gentamycin, 50% (24/48) for ciprofloxacin, 43.75% (21/48) for tetracycline, 35.42% (17/48) for tobramycin, 35.42% (17/48) for trimethoprim-sulfamethoxazole, and 2.08% (1/48) for ampicillin.

Resistance to one or two classes of AMs was identified in 35.42% (17/48) of clinical isolates of *K. pneumoniae* (Figure 1). Among them, the most common were isolates recovered from the stool samples collected from children, 41.18% (7/17) (Figure 1). The most prevalent AMR profile in these non-MDR *K. pneumoniae* isolates was ampicillin resistance (23.53%, 4/17).

We identified an extensively drug-resistant (XDR) phenotype in four clinical *K. pneumoniae* isolates (8.33%, 4/48), which exhibited the identical profile of full resistance to all AMs tested except for colistin (MIC ≤ 1 µg/mL) and tigecycline (MIC < 0.5 µg/mL) (Figure 1). All XDR *K. pneumoniae* isolates were isolated in 2022 from the urine (3) and stool (1) samples of paediatric patients. Notably, resistance to amikacin was detected only in XDR isolates and absent in other *K. pneumoniae* clinical isolates.

Our results indicated that a substantial proportion of human *K. pneumoniae* isolates in this study were MDR, 56.25% (27 out of 48) (Figure 1). The highest susceptibility rates among MDR isolates were found towards meropenem (96.3%, 26/27) and imipenem (96.3%, 26/27), followed by colistin (88.89%, 24/27, MIC ≤ 1 µg/mL), tigecycline (88.89%, 24/27, MIC ≤ 0.5 µg/mL), cefoxitin (85.19%, 23/27), and amikacin (74.07%, 20/27). The MDR isolates displayed lower susceptibility rates towards azithromycin (55.56%, 15/27, MIC ≤ 16 µg/mL), chloramphenicol (51.85%, 14/27), gentamycin (40.74%, 11/27), cefepime (37.04%, 10/27), tobramycin (33.33%, 9/27), and aztreonam (29.63%, 8/27). Susceptibility rates to other AMs were even lower (Figure 1). We identified five MDR isolates resistant to nine classes of AMs, but their resistance profiles were not identical. The most common among MDR isolates was resistance to eight AM classes (33.33%, 9/27). Three isolates (KpA44, KpA46, and KpA373) had the identical AMR profile (Figure 1).

The extended spectrum beta-lactamase (ESBL)-producer phenotype was identified in 45.83% (22/48) of our *K. pneumoniae* clinical isolates (Figure 1). ESBL production, however, was not detected in the XDR isolates. The rate of ESBL production was 77.78% in MDR isolates (21/27) and 5.88% in non-MDR isolates (1/17). Among the ESBL-producing isolates, the highest susceptibility was towards carbapenems, 95.45% (21/22). Carbapenem resistance was detected only in one ESBL-positive isolate, with intermediate resistance to imipenem and full resistance to meropenem. Resistance to colistin was detected in two ESBL-producing isolates, with an MIC of 2 µg/mL. Three ESBL-producers were resistant to tigecycline, with one isolate having an MIC of 1 µg/mL, while the other two had 2 µg/mL. The ESBL-positive MDR *K. pneumoniae* isolates also demonstrated high levels of resistance towards the 3rd and 4th generation cephalosporins: all of them were resistant to ceftriaxone, and only 4.76% (1/21) and 23.81% (5/21) were susceptible to ceftazidime and cefepime, respectively. However, 85.71% (18/21) of the ESBL-producing MDR isolates were susceptible to cefoxitin, suggesting it could be a therapeutic option for the treatment of ESBL-producing *K. pneumoniae* to limit the use of carbapenems. Susceptibility to other AMs in this group was: amikacin (66.67%, 14/21), azithromycin (57.14%, 12/21, MIC ≤ 16 µg/mL), chloramphenicol (47.62%, 10/21), gentamycin (28.57%, 6/21), tobramycin (23.81%, 5/21), and beta-lactam combination agents such as ampicillin-sulbactam (19.05%, 4/21). All MDR ESBL-producing *K. pneumoniae* isolates were resistant to five or more classes of AMs.

Among the six ESBL-negative MDR isolates, the susceptibility rates to all cephalosporins tested were significantly higher compared to ESBL-producers: 100% to ceftriaxone and ceftazidime (*p* < 0.0001) and 83.33% (5/6) to cefepime (*p* < 0.05) (Appendix A). The ESBL-negative isolates were also significantly more susceptible to gentamycin and ciprofloxacin compared to ESBL-producers: 83.33% vs. 28.57% and 66.67% vs. 14.29%, respectively (*p* < 0.05). In addition, one isolate with an MIC to colistin of 2 µg/mL was identified among the MDR ESBL-negative isolates. MDR phenotype with resistance to five or more classes of AMs was significantly lower in ESBL-negative MDR isolates compared to ESBL-producers (50% vs. 100%, *p* < 0.01).

Thus, we detected an alarming presence of carbapenem resistance in XDR *K. pneumoniae* clinical isolates, as well as the high rates of complete resistance to the 3rd and 4th generation cephalosporins. All ESBL-producers were resistant to five or more classes of AMs, hence seriously limiting therapeutic options.

### 2.2. In Vitro Activity of Bacteriophage Preparations Against Human K. pneumoniae Isolates

One of the alternative options for treatment of XDR and MDR infections is phage therapy. We therefore tested our *K. pneumoniae* isolates for susceptibility to the commercial bacteriophage preparations “Bacteriophage *Klebsiella pneumoniae* Purified” (BKpP) and “Bacteriophage *Klebsiella* Polyvalent Purified” (BKPP) (manufacturer: SPA “Microgen” Moscow, Russia), which include phage cocktails active against *K. pneumoniae* and *Klebsiella* spp., respectively.

Complete resistance to both phage cocktails was detected in 27.08% (13/48) of our isolates (Figure 1). Susceptible or intermediate susceptible phenotypes toward BKpP were found in 35 isolates (72.92%) and towards BKPP—in 32 (66.67%). Notably, all XDR *K. pneumoniae* isolates were highly susceptible to BKpP and intermediately susceptible to BKPP. Both phage preparations demonstrated a higher activity against MDR isolates compared to non-MDR isolates: 81.48% (22/27) vs. 47.06% (8/17) for BKpP and 74.07% (20/27) vs. 52.94% (9/17) for BKPP. Thus, the combination of both phage cocktails provided a 100% coverage against XDR *K. pneumoniae* isolates, while the efficiency against the MDR strains was in the range of 74.07–81.48%. At the same time, the three *K. pneumoniae* clinical isolates with the hypermucoviscous (HMV) phenotype (see below) and highly susceptible to antibiotics were mostly resistant against the phage cocktails (Figure 1).

Thus, the commercial phage formulations BKpP and BKPP displayed significant in vitro activity against MDR and XDR *K. pneumoniae* clinical isolates and, therefore, may serve as alternative or adjunct therapies to control this infection.

### 2.3. Hypermucoviscous K. pneumoniae Clinical Isolates

We detected three clinical *K. pneumoniae* isolates (6.25%, 3/48) with a hypermucoviscous (HMV) phenotype (Figure 1) using the string test [12]. All these isolates were recovered from urine samples of adults.

The HMV isolates showed the distinctive AMR profiles of resistance to two classes of AMs and were classified as non-MDR isolates. In particular, all HMV isolates showed resistance to tobramycin (intermediate) and ampicillin (intermediate or resistant), whereas two of them exhibited intermediate resistance to piperacillin-tazobactam (KpA687, KpA704) and one isolate was intermediately resistant to gentamycin (KpA828). In addition, one HMV *K. pneumoniae* clinical isolate, KpA828, was an ESBL-producer. It was the only ESBL-positive isolate among the non-MDR *K. pneumoniae* isolates. No HMV phenotypes were present among the MDR or XDR strains.

### 2.4. Enterobacterial Repetitive Intergenic Consensus (ERIC-PCR) Typing of K. pneumoniae Clinical Isolates

To explore the genetic relatedness among our 48 clinical isolates of *K. pneumoniae*, we used ERIC-PCR (Figure 1). This analysis yielded differential patterns consisting of 9–18 bands. The dendrogram generated from ERIC-PCR data demonstrated that all isolates in our collection showed at least a 63.6% similarity of the band patterns and were grouped into two main clades, A and B (Figure 1). The largest clade, A, composed of 83.33% (40/48) of the isolates, including all four XDR isolates and the majority of MDR isolates (25 out of 27). The non-MDR isolates were dispersed across the tree; however, their prevalence varied from 27.5% (11/40) in the clade A to 75% (6/8) in the clade B. In total, 42 different ERIC types were identified, with 37 unique types, suggesting a significant genetic diversity among our *K. pneumoniae* clinical isolates.

Three out of four XDR isolates displayed a 100% similarity of band pattern (cluster I), suggesting that they may belong to a single clone, while another XDR isolate showed 97.2% similarity to the XDR cluster (Figure 1). In additon, five MDR isolates (including two isolates in cluster II) showed 91.4% similarity to the cluster of XDR isolates. Among other clusters, cluster III grouped together two MDR isolates displaying resistance to nine classes of AMs, while another isolate resistant to nine classes showed 97.2% similarity to the cluster isolates. In addition, two MDR isolates in cluster IV recovered from endotracheal tubes in paediatric patients are of note. Remarkably, all isolates that were clustered together were isolated in the same hospital within the period of up to three weeks, suggesting a possible nosocomial infection.

Thus, ERIC-PCR typing revealed a high genetic diversity among our *K. pneumoniae* clinical isolates, indicating a predominantly polyclonal distribution of *K. pneumoniae* strains. At the same time, there are indications of clonal spread of XDR and some of the MDR isolates.

### 2.5. Whole Genome Sequencing (WGS) of K. pneumoniae Clinical Isolates

A total of 21 *K. pneumoniae* isolates from Armenia were subjected to WGS. This analysis included all four XDR isolates, 13 MDR isolates resistant to seven or more classes of AMs, and one MDR isolate resistant to five classes of AMs. The MDR isolates were selected for WGS based on AMR profile, specimen origin, and year of isolation. In addition, two non-MDR isolates exhibiting HMV phenotype and one isolate susceptible to all AMs tested (except for tobramycin) were also subjected to WGS. ERIC-PCR results were used to avoid redundant sequencing of clonal isolates. Genome sequences are available in the NCBI database under Bioproject PRJNA1141898. Individual accession numbers are listed in Appendix A.

The general genomic information of our *K. pneumoniae* isolates was generated by using the BIGSdb-Pasteur database (https://bigsdb.pasteur.fr/klebsiella/, accessed on 20 September 2024) and the Pathogenwatch resource (https://pathogen.watch, accessed on 20 September 2024). This information is summarised in Appendix A.

The general genomic features, such as genome sizes of *K. pneumoniae* isolates, were in the range of 4.97–5.94 Mb, with GC-content in the range of 56.53–57.76%, which is consistent with the accepted criteria for *Klebsiella* spp. in the BIGSdb-Pasteur database.

### 2.6. Molecular Epidemiology of K. pneumoniae Clinical Isolates

The WGS results indicated that our *Klebsiella* isolates belong to the phylogroup Kp1, *K. pneumoniae sensu stricto*.

A total of 12 different sequence types (ST) were identified among our 21 *K. pneumoniae* clinical isolates (Table 1). The most common was ST395 detected in 7 *K. pneumoniae* isolates (33.33%, 7/21) that were isolated in 2022 (6) and 2024 (1) from paediatric patients. Among the ST395 *K. pneumoniae* isolates were all four XDR isolates with complete resistance to carbapenems and three MDR isolates with resistance to eight classes of AMs. The ST395 isolates were recovered from various sources: urine (3), stool (2), endotracheal tube (1), and wound fluid (1). ST395 is an international high-risk clonal lineage associated with MDR phenotypes of clinical relevance, including the production of carbapenemases and ESBL, as well as resistance to other classes of AMs [25]. The carbapenem-resistant KpA699 isolate, which was recovered from the urine sample of an adult patient in 2022 and exhibited resistance to nine classes of AMs, was assigned to ST15. The other two isolates, KpA13 and KpA230, which were isolated from the throat infection of paediatric patients and resistant to nine classes of AMs, were assigned to *K. pneumoniae* ST39. Two strains resistant to eight classes of AMs were assigned to ST307: KpA500, isolated from the stool sample of an adult in 2018, and KpA204, isolated from the throat infection of a paediatric patient in 2024. In addition, two isolates, KpA250 and KpA314, which were isolated from endotracheal tubes of children in 2022, belonged to ST29.

All other STs were represented by a single isolate, suggesting a significant level of genetic diversity among the clinical isolates of *K. pneumoniae* and confirming our earlier observations with ERIC-PCR. The colistin-resistant KpA511 with resistance to nine classes of AMs was assigned to ST219. The KpA6101 isolate, resistant to eight classes of AMs (including polimyxins) but susceptible to all cephems, was assigned to ST5275. Another colistin non-susceptible isolate, KpA324, with resistance to eight classes of AMs, was the representative of ST449. The remaining MDR isolate subjected to WGS, KpA7002, was assigned to ST873. Among the three non-MDR isolates, the two HMV isolates were assigned to ST107 (KpA704) and ST25 (KpA828, ESBL-producer), while the susceptible isolate to all but one AMs was assigned to ST1480 (KpA857). Thus, despite the limitations of this work due to the non-consecutive collection of *K. pneumoniae* clinical isolates and the small number of genomes sequenced, our results indicate the circulation of international high-risk *K. pneumoniae* clones in Armenia.

Notably, all our ST395 isolates of *K. pneumoniae* belonged to the same sublineage and clonal group (SL395 and CG395, correspondingly) and also shared other important characteristics. An identical core genome sequence type (cgST) was identified among the *K. pneumoniae* ST395 isolates only, whereas all other isolates were assigned to the individual cgSTs (Table 1). In particular, three XDR *K. pneumoniae* ST395 isolates were assigned to cgST-*a23d. These isolates (KpA278, KpA285, and KpA542) were recovered from paediatric patients at the same facility within a three-week period. In addition, one XDR isolate, KpA481, isolated in 2022, was assigned to cgST-*72e4, together with the MDR KpA44 strain isolated in 2024. The limited genetic diversity among the ST395 isolates (Figure 2) is possibly due to the sampling bias, with three isolates obtained from the same facility within a short period of time. Isolates in other *K. pneumoniae* STs were also identified to the sublineage and clonal group levels (Table 1). Isolates within the same ST were assigned to the same sublineages and clonal groups. Furthermore, ST1480 (KpA857) and ST5275 (MDR KpA6101) isolates were assigned to the same sublineage, SL37, whereas ST107 (KpA704, HMV) and ST219 (MDR KpA511) isolates were assigned to the sublineage SL107.

### 2.7. Capsule (K) and Lipopolysaccharide (O) Types Deduced from WGS Data

The most common capsular serotype was K2 (38.1%, 8/21), which was identified in all seven ST395 isolates and in one ST25 isolate, KpA828, with the HMV phenotype (Table 1, sourced from Kaptive [26]). The second most common serotype was K19 (14.29%, 3/21), detected in two ST29 isolates (KpA250, KpA314) and one ST15 isolate (KpA699). Other capsular serotypes were K62 (9.52%, 2/21), identified in two ST39 isolates (KpA13 and KpA204), and KL102 (9.52%, 2/21), detected in ST307 isolates (KpA204 and KpA500). All other capsular serotypes were represented by a single isolate: K10 was detected in the ST107 isolate (KpA704 with the HMV phenotype), K22—in the ST449 isolate (KpA324), K23—in the ST5275 isolate (KpA6101), K39—in the ST1480 isolate (KpA857), K52—in the ST873 isolate (KpA7002), and KL114—in the ST219 isolate (KpA511).

The most common O serotype was O2 (47.62%, 10/21), identified in all ST395 (subtype O2a) and ST307 (subtypes O2a and O2afg) isolates, as well as in the ST5275 isolate (O2afg). The second most common serotype was O1, detected in eight isolates (38.1%) assigned to the following STs: ST15, ST29, ST39, ST107, ST219, and ST449. Collectively, these two O types accounted for 85.71% (18/21) of all sequenced isolates. In addition, O3 serotype (subtype O3b) was identified in the ST1480 isolate KpA857. In the remaining two isolates, the OL101 locus in the ST873 isolate and the unknown (O3/O3a) locus in theST25 isolate were identified, with no recognised O serotype information [26].

### 2.8. Resistome Analysis

The genetic background for the AMR phenotypes observed was explored with the use of WGS data (Appendix A). In this analysis, we mainly focused on determining the genetic basis for AMR in the two previously defined resistance phenotypes, that is, XDR and MDR isolates of *K. pneumoniae*. The emphasis is also made on resistance mechanisms towards AMs of clinical relevance, such as beta-lactams, aminoglycosides, and macrolides, although the others are also described if appropriate.

#### 2.8.1. Resistome Analysis of XDR Isolates of *K. pneumoniae*

Four XDR ST395 isolates with complete resistance to all beta-lactams demonstrated the presence of the metallo-β-lactamase-encoding *bla*_NDM-1_ gene (conferring resistance to carbapenems [27]) and were associated with the *ble*_MBL_ gene (encoding bleomycin resistance protein [28]). The *bla*_NDM-1_ gene, however, was not detected in other clinical isolates (Figure 2). In addition, an identical combination of 5 other genes associated with resistance to beta-lactams (*bla*_CTX-M-15_, *bla*_OXA-1_, *bla*_SHV-11_, *bla*_TEM-1_, and *ftsI* (D350N, S357N)) was found in three XDR ST395 isolates, whereas one isolate (KpA542) lacked the *bla*_CTX-M-15_ gene in this combination. Notably, the ESBL-producer phenotype was not detected in the XDR isolates, which can be explained by the production of the NDM-type carbapenemase that masks this phenotype and confers complete resistance to nearly all beta-lactams, including carbapenems [29].

Complete resistance to amikacin in the XDR *K. pneumoniae* ST395 isolates was associated with the *armA* gene, which encodes 16S rRNA methyltransferase [30], and by the presence of other genes conferring resistance to aminoglycosides such as *aac*(*3*)-*IIe*, *aac*(*6*′)-*Ib*-*cr6*, and *aph*(*3*′)-VIa (Figure 2). Notably, the *armA* gene was detected in the XDR isolates only but not in other isolates.

The high level of resistance to azithromycin in all XDR isolates could be explained by the carriage of the four-gene complex: *mphA*, *mphE*, *mrx*, and *msrE* (Figure 2).

The genetic basis for other mechanisms of AMR among XDR isolates was also explored. Fluoroquinolone resistance in these isolates included the identical combination of *aac*(*6*′)-*Ib*-*cr6*, *gyrA* (S83I), *parC* (S80I), and *qnrS1*, except for the KpA542 isolate, which missed the *aac*(*6*′)-*Ib*-*cr6* gene (Figure 2). Resistance to folate pathway antagonists was associated with the combination of two genes, *dfrA5* and *sul1*, whereas the additional *dfrA1* and *sul2* genes were found in two and three XDR isolates, respectively. Furthermore, the *tet*(A) gene encoding resistance to tetracyclines and the combination of the *catA1* and *catB3* genes conferring resistance to phenicols were detected in the XDR isolates. To the best of our knowledge, this is the first report of NDM-1 carbapenemase-producing and pan-aminoglycoside-resistant human *K. pneumoniae* isolates exhibiting the XDR phenotype/genotype from Armenia. Thus, the extensive bioinformatic analysis of WGS data revealed a highly similar genetic background of the resistome of XDR strains, which encodes the phenotypic resistance to 10 classes of AMs.

#### 2.8.2. Resistome Analysis of MDR Isolates of *K. pneumoniae*

Similarly to XDR strains, in three MDR *K. pneumoniae* ST395 isolates with resistance to 8 classes of AMs, the combination of five genes encoding resistance to beta-lactams (*bla*_CTX-M-15_, *bla*_OXA-1_, *bla*_SHV-11_, *bla*_TEM-1_, and *ftsI* (D350N, S357N)) was also found. These MDR isolates were ESBL-producers and resistant to all beta-lactams tested, excluding carbapenems and cefoxitin. A similar combination of five genes, except for MLST-associated variation in the *bla*_SHV_ gene, was found in isolates belonging to ST29 (2 strains) and ST307 (2 strains). The ST29 isolates were resistant to beta-lactams, whereas one of the ST307 isolates, KpA500, had intermediate resistance to amoxicillin-clavulanic acid, and KpA204 had intermediate resistance to cefoxitin but was susceptible to cefepime. Interestingly, the following combination of four genes was identified in the KpA699 isolate (ST15) with complete resistance to all beta-lactams, including meropenem and cefoxitin, except for intermediate resistance to imipenem: *bla*_CTX-M-15_, *bla*_SHV-28_, *bla*_TEM-1_, and *ftsI* (D350N, S357N). KpA699 had no carbapenemase or AmpC-type beta-lactamase genes, suggesting that other mechanisms must be responsible for this clinically significant phenotype. In all other MDR isolates with resistance to beta-lactams, a combination of two or three beta-lactamase-encoding genes and the *ftsI* (D350N, S357N) gene was identified (Figure 2). MDR isolates possessed *bla*_CTX-M_, which was not detected only in one isolate (KpA6101, ST5275). KpA6101 had a combination of the *bla*_LAP-2_, *bla*_SHV-11_, *bla*_TEM-1,_ and *ftsI* (D350N, S357N) genes, was ESBL-negative, and was susceptible to all cephems. In the KpA324 isolate, the combination of the *bla*_CTX-M-14_, *bla*_SHV-33_, and *ftsI* (D350N, S357N) genes was associated with susceptibility to ampicillin-sulbactam and intermediate resistance to ceftazidime and amoxicillin-clavulanic acid. The MDR isolates also possessed *bla*_SHV_, which was absent in one isolate only (KpA230, ST39). KpA230 had the *bla*_CTX-M-15_, *bla*_TEM-1_, and *ftsI* (D350N, S357N) genes in combination and displayed complete resistance to all beta-lactam combination agents and cephems, except for cefepime.

Regarding aminoglycoside resistance determinants, all but one MDR *K. pneumoniae* isolate (13 out of 14) carried aminoglycoside-modifying enzyme genes in combination (Figure 2). The most common genes were *aph*(*3*′′)-*Ib* (71.43%, 10/14), *aph*(*6*)-*Id* (64.29%, 9/14), *aac*(*3*)-*IIe* (64.29%, 9/14), and *aac*(*6*′)-*Ib*-*cr6* (50%, 7/14). In all isolates showing full resistance to gentamicin, the *aac*(*3*)-*IIe* (9 isolates) or *aac*(*3*)-*IId* (2 isolates) genes were identified. These genes encode aminoglycoside 3-N-acetyltransferase enzymes, which inactivate gentamicin and tobramycin. In gentamicin-susceptible isolates, these genes were not detected (*p* < 0.01).

The genetic basis for amikacin resistance in five MDR isolates was more complex. The *aph*(*3*′)-VIa gene conferring resistance to amikacin [31], in combination with the *aac*(*6*′)-*Ib*-*cr6* and *aac*(*3*)-*IIe* genes, was identified in one isolate, KpA44 (Figure 2). There were two gene profiles encoding for aminoglycoside-modifying enzymes. The first was represented by the *aac*(*3*)-*IIe* and *aac*(*6*′)-*Ib*-*cr6* genes and was detected in KpA7001. These two genes, in combination with the additional *aph*(*3*′′)-*Ib* and *aph*(*6*)-*Id* genes, were detected in KpA250 and KpA314. These gene profiles were also detected in amikacin-susceptible isolates (KpA769 and KpA204). Finally, the combination of *aac*(*3*)-*IIe*, *aph*(*3*′′)-*Ib*, and *aph*(*6*)-*Id* genes was identified in one isolate, KpA7002, with intermediate resistance to amikacin.

Among the MDR isolates with azithromycin resistance (5 isolates, MIC ≥ 64 µg/mL), three isolates had the combination of the *mphA* and *mrx* genes (Figure 2). In another isolate with an MIC of ≥ 64 µg/mL, only one gene, *mphA*, was detected. Macrolide resistance genes were not identified in one isolate (KpA500, MIC ≥ 64 µg/mL), suggesting that other mechanisms are involved in azithromycin resistance. In addition, the single *mphE* gene was detected in one isolate, KpA699, that had an MIC < 16 µg/mL to azithromycin.

The mechanisms of resistance to other classes of antimicrobials in the clinical MDR *K. pneumoniae* isolates were also explored. Resistance to phenicols in these isolates was associated with the *cat2* gene, as well as with the combinations of the *catA1* gene with the *floR* or *catB3* genes (Figure 2). The presence of a single *catB3* gene was detected in one chloramphenicol-resistant isolate, KpA204. This gene, however, was also detected in three chloramphenicol susceptible isolates (KpA250, KpA314, and KpA500). In one isolate with full resistance to chloramphenicol (KpA324), no known acquired phenicol resistance gene can be found.

Resistance to ciprofloxacin in these isolates was commonly associated with the combination of mutations in both the *gyrA* (S83I or S83F) and *parC* (S83I) genes with the *qnr* genes (*qnrS1*, *qnrB1*, *qnrS1*, and *qnrB20*). In one isolate, KpA699, the additional *aac*(*6*′)-*Ib*-*cr6* gene, as well as the presence of the two substitutions in the *gyrA* (S83F, D87A) gene, were detected. In the other three ciprofloxacin-resistant isolates, the *qnrB1* and *aac*(*6*′)-*Ib*-*cr6* genes in combination (two ST29 isolates) or the single *qnrB1* gene (KpA511) were identified. Furthermore, the combination of *qnrB20* and *qnrS1* genes was detected in the ciprofloxacin-resistant isolate KpA230, whereas a single *qnrS1* gene was associated with the intermediate resistance phenotype of KpA6101.

The presence of the *tet*(A) and *tetR*(A) genes was detected in all isolates exhibiting resistance to tetracycline, except for one isolate (KpA511), which carried the *tet*(B) and *tetR*(B) genes. In the only isolate showing an intermediate phenotype to tetracycline (KpA699), the *tet* genes were not detected. Notably, the three tigecycline non-susceptible isolates (KpA13, KpA204, and KpA7001) harboured the same *tet*(A) gene variant as the tigecycline-susceptible isolates. However, no acquired genetic determinants associated with resistance to this antibiotic were detected. The high level of resistance to folate pathway antagonists in MDR isolates (100%) was in agreement with the presence of the *dfrA* gene variants (*dfrA1*, *dfrA5*, *dfrA14*, *dfrA17*, *dfrA12*, and *dfrA27*), in combination with the *sul1* or/and *sul2* genes. The *fosA6* and *uhpT* (E350Q) genes encoding resistance to fosfomycin were detected in all MDR isolates, as well as the *arnT*, *eptB*, and *ompA* genes conferring resistance to peptide antibiotics. In addition, the *aar*-*3* gene encoding resistance to ansamycins was present in one isolate, KpA699. Of note, any acquired resistance determinants associated with resistance to colistin were not identified in our MDR isolates.

#### 2.8.3. Resistome Analysis of Non-MDR Isolates of *K. pneumoniae*

The presence of AMR determinants in the three non-MDR isolates was also examined. The only AMR mechanisms encountered were beta-lactam resistance determinants (Figure 2). KpA857 isolate, susceptible to all beta-lactams, possessed *ftsI* (S357N, D350N). Two non-MDR HMV isolates carried the identical combination of two genes, *bla*_SHV-11_ and *ftsI* (D350N, S357N), but with different phenotypes. KpA704 was ESBL-negative and had intermediate resistance to ampicillin and piperacillin-tazobactam, while KpA828 was an ESBL-producer with full resistance to ampicillin.

### 2.9. Efflux Systems in K. pneumoniae Clinical Isolates

The main multidrug efflux systems (ES) in *Klebsiella* spp., AcrAB [32] and OqxAB [33], were identified in 100% and 95.24% of *K. pneumoniae* isolates, respectively (Figure 2). The *oqxA* and *oqxB* genes were not detected in one isolate only, the MDR KpA13 isolate (ST39). In addition, the following most prevalent ES were identified: AcrEF (100%), EefAB (100%), EmrAB (100%), KpnEF (100%), KpnGH (95.24%, 20/21), LptD (100%), MacAB (100%), and MsbA (100%). The prevalence of other efflux pumps was lower: Tet(A) (71.43%, 15/21), Tet(B) (4.76%, 1/21), QacEdelta1 (52.38%, 11/21), FloR (9.52%, 2/21), and CrcB (4.76%, 1/21).

The *acrR* gene encoding for the repressor of the AcrAB-TolC pump [34] was present in all genomes. However, in five MDR isolates (KpA13, KpA230, KpA44, KpA6101, and KpA699), an identical profile of substitutions in this gene was identified: P161R, G164A, F172S, R173G, L195V, F197I, K201M (sourced from the ResFinder database [35,36], http://genepi.food.dtu.dk/resfinder, accessed on 20 September 2024). Mutations in the *acrR* gene contribute to the overexpression of the AcrAB-TolC complex, leading to the higher level of resistance to multiple antibiotics [32]. In addition, the *marA* gene that encodes the global activator MarA mediating the overexpression of the AcrAB pump [34] was identified in all our isolates. Also, mutations in the *marR* gene encoding MarR repressor of *marA*, which also lead to overexpression of the AcrAB pump and a reduced susceptibility to multiple antibiotics [34], were detected. The prevalence of *ramA* and *ramR* genes that also encode the regulators of the AcrAB pump (activator of *acrAB* and repressor of *ramA*, correspondingly [34]) was lower, 66.67% (14/21). It should be noted here that the *ramAR* genes were not detected only in the ST395 isolates. In addition, in isolate KpA699, a substitution in the *ramR* (A19V) gene was detected. This mutation is known as contributing to a reduced susceptibility to tigecycline [37]; the isolate, however, was susceptible to this antibiotic. Regarding other genes involved in efflux pump regulation, the following genes were identified in all our isolates, irrespective of AM resistance phenotype: *baeR*, *crp*, *leuO*, *h*-*ns*, and *rsmA*. Of note, the *rarA* gene encoding a transcriptional activator of the efflux pump OqxAB [37] was not detected only in KpA13, which was also negative for *oqxAB* genes. Futhermore, the *emrR* gene, a negative regulator of the EmrAB-TolC efflux system associated with resistance to nalidixic acid and thiolactomycin [38], was not detected only in non-MDR HMV KpA828, with potential overexpression of the EmrAB pump. In addition, the *tet*(R) repressor gene (76.19%) was present in all isolates carrying Tet pumps.

The results indicated a common set of predominant *Klebsiella* pumps in most isolates, regardless of their AMR phenotype. More differences were identified in the repertoire of genes regulating efflux pumps. In particular, the absence of *ramAR* genes in all our *K. pneumoniae* isolates belonging to the ST395 is of note. Additionally, the co-occurrence of mutations in the *acrR* and *ramR* genes was detected in one isolate only, KpA699, for which the resistance mechanisms to three classes/subclasses of AMs (carbapenems, cephamycins, and tetracyclines) cannot be explained by the presence of any acquired resistance genes. Potentially, the overexpression of the AcrAB efflux pump due to mutations in the *acrR* and *ramR* genes may contribute to the aforementioned resistances in this isolate.

### 2.10. Analysis of ompK Genes

Loss of, or mutations in, the major porins of *K. pneumoniae* result in AMR, and this possibility was explored among our isolates. The intact *ompK35*, *ompK36*, and *ompK37* genes were identified in 42.86% (9/21) of the sequenced *K. pneumoniae* clinical isolates, including all non-MDR isolates (3) and 33.33% of the MDR isolates (6/18).

The *ompK35* gene (GenBank: AJ011501.1) was identified in 12 isolates (57.14%) with 98.43–100% identity, whereas in the remaining nine isolates (42.86%), a truncated form of OmpK35 porin was predicted. It should be noted here that an identical deletion in the *ompK35* gene leading to the truncated porin protein was predicted in all ST395 isolates (four XDR and three MDR). In addition, in the KpA500 (ST307) isolate, an insertion of 26 bp in the *ompK35* gene after nucleotide 226 resulted in a premature stop codon, which also resulted in a truncated protein form. In the KpA699 isolate, a premature TAA stop codon in the *ompK35* gene appeared as the result of point mutation, also leading to a truncated form of the protein. In this isolate and in one ST395 isolate (KpA44), OmpK35 porin deficiency was coupled with mutations in the *ompK36* and *ompK37* genes (see below). In all other isolates with the truncated form of OmpK35, the intact *ompK36* and *ompK37* genes were detected.

It should be noted here that the *ompK36* (GenBank: Z33506.1) and *ompK37* (GenBank: AJ011502.1) genes were identified in all isolates, in five of which (23.81%) mutations in both of these genes were observed. These five isolates (KpA13, KpA44, KpA230, KpA6101, and KpA699) had the identical combination of nine mutations in the *ompK36* gene (N49S, L59V, G189T, F198Y, F207Y, A217S, T222L, D223G, and N304E; sourced from the ResFinder database [35,36]). A217S substitution is associated with resistance to carbapenems, while the other mutations are associated with cephalosporin resistance. In KpA699, these mutations were combined with the Gly134Asp135 duplication in loop 3 of the OmpK36 protein (OmpK36GD, sourced from Kleborate [39]), which leads to the attenuated diffusion of carbapenems [40]. An identical combination of the four mutations in the *ompK37* gene (I70M, I128M, N230G, and E244D; sourced from the ResFinder database [35,36]) was found in three isolates: both ST39 isolates (KpA13 and KpA230) and one ST395 isolate (KpA44). All these mutations are known as associated with resistance to carbapenems [35,36]. In addition, the combination of two mutations (I70M and I128M) was found in KpA6101 (ST5275) and KpA699 (ST15). Notably, among these five isolates with mutations in the *ompK36* and *ompK37* genes that are associated with resistance to carbapenems, only one isolate, KpA699, was non-susceptible to carbapenems (intermediate to imipenem and resistant to meropenem), whereas other isolates were sensitive.

Thus, mutations in genes encoding OmpK porins may contribute to the reduced permeability to AMs. The combined mutations in the *ompK36* and *ompK37* genes (23.81%, 5/21) and/or a truncated OmpK35 porin (42.86%, 9/21) may play a part in the increased resistance towards clinically important AMs in 57.14% (12/21) of isolates. In particular, these mechanisms may be responsible for resistance to carbapenems in carbapenemase-negative strains such as KpA699. It carries mutations in the *ompK36* (OmpK36GD, A217S) and *ompK37* (I70M, I128M) genes that are associated with resistance to carbapenems, as well as a truncated form of OmpK35 porin.

### 2.11. Virulence-Related Genes

The most prevalent (13/21, 61.9%) was the virulence score of 1 (sourced from Kleborate [39]), which included isolates carrying the *ybt* locus encoding iron-scavenging siderophore yersiniabactin [41] (Table 2). Among these isolates, the non-MDR HMV KpA828 (ST25) strain is of note. In all but one of these isolates, the *ybt* loci (*ybt* lineages 1, 8, 9, 14, 15, and 16) were located within various structural variants of integrative conjugative elements (ICE*Kp4*, ICE*Kp9*, ICE*Kp3*, ICE*Kp5*, and ICE*Kp12*) and were mainly distributed according to STs (Table 2). In one AM-susceptible isolate, KpA857 (ST1480), however, *ybt* (*ybt* 4) was located on the pCAV1099-114 plasmid (IncFIB(K) incompatibility group).

The virulence score of 4 was assigned by Kleborate to five isolates (23.81%, 5/21) that belonged to ST395 (Table 2). Four of them were carbapenem-resistant XDR strains and one MDR strain (KpA7001). In all of these isolates, the virulence determinants were associated with ICE*Kp12* and included: the *ybt* locus (lineage 16, sequence type: 53–2LV), the *iucABCD* and *iutA* genes encoding aerobactin siderophore [47] (*iuc* 1; AbST: 63), and the *rmpA2* gene (allele 28; sourced from the BIGSdb-Kp database) encoding the regulator of mucoid phenotype [48]. In addition, in one isolate (KpA481), the additional *rmpADC* (*rmp* 1/KpVP-1 lineage) and *peg344* (metabolic transporter of unknown function) genes were identified. A frameshift mutation in the *rmpA2* gene due to insertion within a poly-G tract, which results in a premature stop codon (TAA) and truncated protein (47%), was identified in all *rmpA2*-positive isolates. This mutation may explain the HMV-negative phenotypes of these isolates found earlier by the sting-test. Despite the HMV-negative phenotypes, however, the presence of other virulence markers indicates a high virulence potential. In all five isolates, the *rmp* and aerobactin-encoding loci were co-localised on the same contigs. In addition, these isolates shared an identical plasmid replicon profile with the characteristic presence of IncFIB(K)/IncFIB(pNDM-Mar)/IncHI1B(pNDM-MAR) replicons, except for the KpA481 isolate that had only the IncFIB(pNDM-Mar) replicon.

The virulence score of “0” was assigned to three *K. pneumoniae* clinical isolates (14.29%), one of which was the non-MDR KpA704 (ST107) isolate with the HMV phenotype, while the other two isolates were MDR, KpA204 (ST307) and KpA6101 (ST5275).

In addition to Kleborate, the presence of virulence factors in WGS data was further analysed with VFanalyzer [49] and the BIGSdb-Kp database at the Pasteur Institute. According to VFanalyzer, the genes encoding Type 1 and Type 3 fimbriae were detectable in all our isolates, except for the *mrkH* gene (transcriptional activator regulating biofilm formation [50]), which was missing in five isolates belonging to ST395 (Appendix A). Other genes identified in all of the isolates were *ent* locus encoding siderophore enterobactin, the K locus that determines polysaccharide capsule type, the *csrAB* genes involved in capsule synthesis regulation, the chromosome-located *iro* genes encoding siderophore esterase (*iroE*) and salmochelin receptor (*iroN*), and the chromosomal *iutA* gene for ferric aerobactin receptor.

The three almost entire clusters of Type VI Secretion System (T6SS-I, T6SS-II, and T6SS-III), but with missing genes encoding Tli1, LysM, and two hypothetical proteins, were identified only in two ST307 isolates (Appendix A). The T6SS-I and T6SS-III clusters were detectable in almost all our isolates (95.24% and 100%, respectively), whereas the T6SS-II cluster was not identified in 61.9% (13/21) of isolates. The intact T6SS-I cluster similar to that in the MDR *K. pneumoniae* HS11286 strain (NC_016845) was identified in seven isolates (33.33%). Six of them belonged to ST395 and one, KpA324, to ST449. The remaining 14 isolates lacked the *tle1* and/or *tli1* genes (effector-immunity pair of proteins participating in intra- and interspecies antagonism [51]) (Appendix A). This pair of toxin-antitoxin genes was absent in seven isolates (33.33%) belonging to ST15, ST25, ST107, ST219, ST873, ST1480, and ST5275. The single *tle1* gene was detected in two ST29 isolates (9.52%), and the single *tli1* gene was found in five isolates (23.81%) belonging to ST39, ST307, and ST395. Notably, the *tli1* gene copy number ranged from one in ST39 isolates to seven in ST307 isolates. In one isolate, KpA828 (ST25), a reduced T6SS-I cluster limited to only two genes (*clpV*/*tssH* and *hcp*/*tssD*) was found. Other virulence-related genes were detected at lower frequencies. However, the presence of the following fimbrial adherence determinants is of note: the *stbABCDE* genes identified in two ST29 isolates (9.52%) and the *steB* and *stfD* genes detected in two ST307 isolates (9.52%).

### 2.12. In Silico Identified Plasmid Replicons

Plasmid replicons were identified in almost all sequenced isolates, except for one isolate, KpA828 (non-MDR, HMV), which also had the smallest genome size among our isolates (Appendix A). The number of plasmid replicons per isolate ranged from 1 to 7 (Table 2).

Notably, a high number of plasmid replicons was characteristic for the isolates belonging to ST395 (from 4 to 7 replicons). The highest number of plasmid replicons was detected in four ST395 isolates, three of which were XDR strains carrying the *bla*_NDM-1_ gene (KpA278, KpA285, and KpA542) and one—the MDR isolate KpA7001. In these isolates, an identical profile of the following 7 plasmid replicons was identified: Col(pHAD28) (KU674895), ColRNAI (DQ298019), IncFIB(K) (JN233704), IncFIB(pNDM-Mar) (JN420336), IncFII(K) (JN233704), IncHI1B(pNDM-MAR) (JN420336), and IncR (DQ449578). In another XDR isolate carrying the *bla*_NDM-1_ gene, KpA481, the plasmid profile was represented by 4 replicons only: Col(pHAD28), ColRNAI, IncFIB(pNDM-Mar), and IncR. The strain also had the smallest genome size among the XDR isolates (Appendix A).

The most common plasmid replicons were IncR and IncFIB(K), detected with the same prevalence of 57.14% (12/21). Among IncR-positive isolates, the most common AMR gene identified on the replicon contigs was the *bla*_CTX-M-15_ gene, which was detected in seven isolates. Notably, the IncR replicon was identified in all seven ST395 isolates. In three of them it was co-located with the *bla*_CTX-M-15_ gene, whereas in two other isolates the replicon was located on the same contigs carrying the MDR regions. In particular, the following genes were identified in KpA769: *bla*_CTX-M-15_, *aac*(*3*)-*IIe*, *catB3*, *bla*_OXA-1_, *aac*(*6*′)-*Ib*-*cr6*, *sul1*, *qacEdelta1*, *dfrA1*, *tetR*(A), and *tet*(A). Of these, five genes were also detected in the KpA7001 contig with the IncR replicon: *bla*_CTX-M-15_, *aac*(*3*)-*IIe*, *catB3*, *bla*_OXA-1_, and *aac*(*6*′)-*Ib*-*cr6*. Besides, the *tet* genes were detected in the IncR contigs of two isolates belonging to other STs: *tet*(B) and *tetR*(B) in KpA511 (ST219) and *tet*(A) and *tetR*(A) in KpA6101 (ST5275). The IncFIB(K) replicon was detected in 12 isolates, but the AMR genes that were co-located with the replicon on the same contigs were found only in two isolates, KpA7002 (ST873) and KpA500 (ST307). In both these isolates, the *aph*(*6*)-*Id*, *aph*(*3*″)-*Ib*, and *sul2* genes were co-located with the IncFIB(K) replicon on the same contigs, while in KpA500 the co-location of four additional AMR genes (*bla*_TEM-1_, *catB3*, *bla*_OXA-1_, and *aac*(*6*′)-*Ib*-cr6) was detected.

Among other plasmid replicons, the prevalence of Col(pHAD28) and ColRNAI plasmid replicons was 42.86% (9/21) and 33.33% (7/21), correspondingly. The combination of these replicons was identified in all seven ST395 isolates, but no linkage with any AMR genes was detectable in the replicon contigs. In KpA324 (ST449), however, the Col(pHAD28) replicon was co-located with the *bla*_CTX-M-14_ gene on the same contig. Notably, the IncFII(K) replicon was identified in eight isolates (38.1%) and always in combination with the IncFIB(K) replicon, while the IncFIB(K) replicon on its own was present in four other isolates. The prevalence of IncFIB(K)(pCAV1099-114) (CP011596) plasmid replicon was lower, 19.05% (4/21); however, in three out of four isolates, the AMR genes were co-located on the replicon contigs. In particular, in the KpA511 (ST219) isolate, a 6588 bp resistance region carrying the following 10 genes was identified: *qnrS1*, *aph*(*6*)-*Id*, *aph* (*3*″)-*Ib*, *sul2*, *dfrA12*, *aadA2*, *qacEdelta1*, *sul1*, *mrx*, and *mphA*. In two isolates belonging to ST39, KpA13 and KpA230, the number of AMR genes on the replicon contig was restricted to four (*qnrS1*, *aph*(*6*)-*Id*, *aph* (*3*″)-*Ib*, and *sul2*) and three (*qnrS1*, *mphA*, and *mrx*), respectively. Notably, the IncFIB(K)(pCAV1099-114) replicon was also detected in the AM sensitive-isolate KpA857 (ST1480). The IncFIA(pBK30683) (KF954760) plasmid replicon was found only in two ST29 isolates (KpA250 and KpA314) and was not associated with AMR genes. The IncQ (M28829) plasmid replicon was identified in one isolate only, KpA44 (ST395), and the *aph*(*3*′)-*IV* gene (amikacin resistance) was co-located with this replicon on the same contig.

### 2.13. Prophage Regions and CRISPR Arrays in *K. pneumoniae* Clinical Isolates

The genomic sequences were analysed for the presence of prophage regions using the PHASTEST web server [43,44,45] (https://phaster.ca, accessed on 20 September 2024). The prophage regions were identified in all our isolates (Table 2). A total of 24 prophage regions were identified, and their main characteristics are summarised in Appendix A. The most prevalent phage was Klebsi_phiKO2 (NC_005857), detected in 38.1% of isolates. The number of prophages per isolate ranged from 1 to 6. Notably, the highest number of prophages was detected in isolates belonging to ST395 (Table 2). All these ST395 isolates, except KpA769, shared an identical profile of the following 6 prophages: Edward_GF_2 (NC_026611), Escher_HK639 (NC_016158), Klebsi_3LV2017 (NC_047817), Klebsi_ST147_VIM1phi7.1 (NC_049451), Klebsi_ST512_KPC3phi13.2 (NC_049452), and Salmon_SEN34 (NC_028699). Interestingly, all these prophages, except for one (Escher_HK639), were detected in ST395 isolates only. The KpA769 isolate was missing two prophages from this profile (Escher_HK639 and Salmon_SEN34) but had the additional two prophages, Escher_RCS47 (NC_042128) and Salmon_Fels_1 (NC_010391). Among the other isolates, an identical prophage profile was detected only in two ST29 isolates, KpA250 and KpA314, which harboured the following three prophages: Escher_RCS47, Klebsi_phiKO2 (NC_005857), and Klebsi_ST15_OXA48phi14.1 (NC_049454).

The prophage regions were analysed for the presence of AMR and virulence-related genes using CARD [52] and VFanalyzer [49] tools. No virulence-associated genes were detected in the prophage regions, consistent with the previously published results demonstrating the low prevalence of virulence factor genes in *K. pneumoniae* prophages [53]. On the contrary, the carriage of AMR genes on prophages was significant, and these genes were detected in nine prophages within the genomes of six *K. pneumoniae* isolates (Table 3). Notably, the prophages carried six genes of resistance to beta-lactams, with one of them, Escher_RCS47, harbouring two of them in the KpA769 (ST 395) isolate.

Interestingly, Escher_RCS47 was also identified in two other isolates belonging to ST29 and ST39 (Table 3). This phage is one of the major phage types harbouring AMR genes, which was found to be predominantly located on plasmids [53,54]. It should be noted here that in KpA769 (ST395), the Escher_RCS47 region was detected on the contig carrying the IncR plasmid replicon and the MDR region including ten AMR genes mentioned above (see Section 2.12). Of these ten genes, eight that encode resistance to 5 classes of AMs and quaternary ammonium compounds were located in the prophage region (Table 3). The role of prophage-located AMR genes in the MDR phenotype is also noticeable for other isolates listed in Table 3. In particular, in the KpA699 isolate (ST15), two Klebsi_phiKO2 phage regions, which are probably the result of superinfection, had, in summary, nine AMR genes conferring resistance to five classes of AMs.

Another phage carrying multiple AMR genes is the Staphy_SPbeta_like phage, which confers resistance to six classes of AMs in the KpA204 (ST307) isolate. Importantly, there are indications that this phage can move beyond the generic barriers and can be detected in several species of Gram-positive and Gram-negative bacteria [53,54]. We performed more detailed analysis of this phage region with MobileElementFinder (https://cge.food.dtu.dk/services/MobileElementFinder, accessed on 20 September 2024) and found the presence of two composite transposons: (i) cn_3826_IS26 (isfinder db, accession X00011) harbouring *catB3*, *bla*OXA-1, and *aac*(*6*′)-*Ib*-*cr* genes and (ii) cn_15047_IS26 (isfinder db, accession X00011) carrying *tet*(A), *tetR*, and *qnrB1* genes. This combination of two mechanisms of horizontal gene transfer may explain its mobility beyond the generic barriers.

Additionally, the AMR genes located in two phage regions in the KpA13 (ST39) isolate encoded resistance to four classes of AMs. These results indicated an important contribution of both intact and questionable prophages to the MDR phenotype in a subset of our clinical *K. pneumoniae* isolates. Of note, we did not detect any genes associated with AMR or virulence-related genes in the incomplete phage regions that were identified in our isolates.

Clustered regularly interspaced short palindromic repeats (CRISPR) arrays were identified in five isolates (23.81%) belonging to the following STs: ST15 (KpA699, 2 arrays), ST39 (KpA13 and KpA230, 1 array), ST449 (KpA324, 2 arrays), and ST873 (KpA7002, 1 array). Of these, the KpA699 isolate carrying two CRISPR arrays had four plasmids and five prophage regions, whereas the low number of phages and plasmid replicons was detected in the other isolate carrying two CRISPR arrays, KpA324 (2 and 1, respectively). The majority of *K. pneumoniae* strains, however, had no detectable CRISPR arrays, thus with no phage immunity traits and, therefore, no boundaries against phage infections. Subsequently, the strains within the problematic ST395 or ST307 are not immune against phages and can easily participate in horizontal gene exchange via phages, thus contributing to their evolution in the form, for example, of the acquisition of AMR genes.

The absence of phage immunity in the XDR and MDR strains belonging to ST395 makes phage therapy a viable option to treat and control this infection.

### 2.14. Phylogenomic Analyses

To estimate genetic relatedness among our *K. pneumoniae* clinical isolates based on genomic sequences, we performed whole genome-based phylogenetic analysis using the type strain genome server (TYGS) (https://tygs.dsmz.de, accessed on 20 September 2024) [55,56]. The results demonstrated an ST-based distribution of the genomic sequences across the phylogenetic tree (Figure 2). All isolates belonging to the same ST were grouped together, with a high similarity score, and the average nucleotide identity (ANI, https://www.ezbiocloud.net/tools/ani, accessed on 20 September 2024) values were in the range of 99.97–99.99%. In particular, *K. pneumoniae* ST395 isolates with the XDR or MDR phenotypes and genotypes were clustered together, displaying a low genetic diversity, with an ANI value of 99.99%. In addition, the isolates representing the same clonal lineage were also grouped together, although with a lower degree of similarity: KpA6101 (ST5275) and KpA857 (ST1480) were assigned to sublineage SL37, while KpA511 (ST219) and KpA704 (ST107) were assigned to sublineage SL107. All other branches were represented by single isolates.

To place our *K. pneumoniae* clinical isolates within the international context, we performed whole genome-based phylogenetic analysis involving our strains and the most closely related genomic sequences from other countries. The latter genomic sequences of *K. pneumoniae* strains were obtained from the Pathogenwatch global resource (https://pathogen.watch, accessed on 20 September 2024). The most closely related strains of *K. pneumoniae* were identified based on core genome analysis, and the combined phylogenetic analysis revealed two main clades, A and B (Figure 3A, Appendix A). Then, the whole-genome comparisons were made with the combined datasets, including our and international strains, using the TYGS resource (Figure 3B).

The noticeable difference between the clades A and B is that the former includes a more diverse range of countries and continents, while the latter is mainly confined to Europe, with minor inclusions from other geographical locations (Figure 3B). Also, the years of isolation of the clade A strains were earlier compared to clade B (2004–2022 vs. 2008–2024). Our seven XDR and MDR *K. pneumoniae* ST395 isolates demonstrated low genetic diversity and were located within clade B (Figure 3B). The most genetically close strains were from Russia and Germany, with the ANI values of 99.96–99.98%.

Another international high-risk clone, ST307, was also located within the clade B (Figure 3B). These MDR *K. pneumoniae* ST307 strains were isolated in the region from 2018 to 2024, including our isolates and also the isolates in another study [24]. The closest genomic matches were the strains from the USA. In particular, four *K. pneumoniae* ST307 clinical isolates from Armenia collected in 2018 and 2019 were close to the strain from the USA collected in 2014, while the KpA204 isolate from 2024 was close to the strains collected in the USA in 2019 (ANI values of 99.97–99.98%). The low genetic diversity was also identified in ST39 strains within clade B (Figure 3B). The ANI value of 99.98% was obtained for the isolates collected in Armenia in 2024 and the strains isolated in Ethiopia in 2020.

The carbapenem-resistant strain KpA699, belonging to the international high-risk clone ST15 and isolated in 2022, was located within the clade A (Figure 3B). The closest genomic matches to this isolate were the strains from Turkey, collected in 2013 and 2014, and the strain from Australia, isolated in 2014 (Figure 3B). The low genetic diversity was also observed among the ST449 strains in clade A, which included the KpA324 isolate from Armenia and strains from Germany, Spain, Madagascar, and Japan (ANI values of 99.97–99.99%). Another MDR isolate from Armenia, KpA511 (ST219), had the closest genomic matches with the strains from India and Turkey (ANI values of 99.98% and 99.99%, correspondingly). The remaining isolates from Armenia had less relatedness to international clones and were clustered within the groups demonstrating a higher level of genomic diversity (Figure 3B). It should be noted here that the KpA6101 isolate in our collection was the only isolate belonging to ST5275 in the Pathogenwatch database. We were unable, therefore, to perform comparative genomic analysis within this ST. In the phylogenetic tree, this single ST5275 genome formed a sister group with the ST1480 genomes (Figure 3B).

## 3. Discussion

The global rise of *K. pneumoniae* pathotypes with hypervirulent and MDR traits poses a significant threat to public health. To deal with this threat, many countries implemented monitoring programs, which help to understand the epidemiology and drug resistance of this pathogen and take the necessary measures to treat and control it. In particular, this information is especially useful for doctors, who deal with severely infected patients and have to make a prompt decision regarding the empirical AM therapy, which should be the most appropriate under the current circumstances. However, in certain regions of the world, there is a paucity of information regarding the local *K. pneumoniae* pathotypes, and this compromises the timely and efficient therapy to reduce excessive morbidity and mortality rates. In particular, this is the case for Armenia, and the main objective of this work was to collect epidemiological and drug resistance data in the interest of public health. Thus, we performed complex, in-depth analysis of *K. pneumoniae* pathotypes circulating in this region.

We analysed the collection of 48 isolates of *K. pneumoniae*, which were obtained from hospital patients during the period from 2018 to 2024. Extensive analysis of AMR with the panel of 22 antibiotics, which covered 12 different classes of antibiotics, revealed that the majority of the isolates are XDR (resistance to at least 10 classes of AMs) and MDR (resistance to at least five classes of AMs) bacteria, comprising together 64.58% of the isolates. Resistance to less than five classes of AMs was encountered in 35.42% of the isolates. Thus, therapeutic options for the majority of *K. pneumoniae* infections are limited. In particular, XDR isolates (8.33%, 4/48) demonstrated an identical profile of complete resistance to 10 classes of AMs, including carbapenems, but they were still susceptible to colistin and tigecycline.

The MDR isolates of *K. pneumoniae* (56.25%, 27/48) demonstrated a high proportion of ESBL-producers, 77.78% (21/27), with resistance to the 3rd and 4th generation cephalosporins. Interestingly, co-resistance to more classes of AMs was mainly correlated with the ESBL production, in contrast to the non-producers. In the MDR group, intermediate resistance to colistin and tigecycline was found in 11.11% (3/27) of the isolates. The highest susceptibility in MDR isolates was to carbapenems and cefoxitin. The latter suggests that cefoxitin, a cephamycin resistant to ESBL hydrolysis, could be a viable therapeutic option for the treatment of infections caused by MDR *K. pneumoniae* to limit the use of carbapenems as has been suggested earlier [57].

The non-MDR isolates demonstrated susceptibility to all first-line drugs used to treat *K. pneumoniae* infections. Interestingly, the HMV phenotype, which is considered to be one of the markers of hypervirulence [58], was identified in three non-MDR isolates (14.29%). However, this phenotype was not detectable among our XDR and MDR isolates.

One of the alternatives to AMs is phage therapy, and for this purpose, we tested two commercial phage preparations, BKpP and BKPP (SPA “Microgen” Moscow, Russia), which include phage lines active against *K. pneumoniae* and *Klebsiella* spp., respectively. Both these phage cocktails displayed significant in vitro activity against *K. pneumoniae* clinical isolates, especially the XDR (100%) and MDR (75–82.14%) strains. Given the limited options of AM therapy against these XDR or MDR isolates, the high efficiency of the commercial bacteriophage preparations could be of interest, suggesting that they may serve as an alternative/adjunct therapy to control these hard-to-treat infections.

The initial genetic characterisation of our isolates was performed with ERIC-PCR fingerprinting, which allows us to estimate genetic relatedness among the isolates of enteric bacteria and vibrios [59]. This analysis demonstrated a high genetic diversity among our *K. pneumoniae* strains, demonstrating their predominantly polyclonal structure. However, the XDR and some of the MDR isolates produced identical or very similar ERIC-PCR fingerprints, suggesting the possibility of the clonal spread. The ERIC-PCR results were also used as one of the criteria for selection of strains for WGS analysis to avoid redundant sequencing.

The selected 21 human *K. pneumoniae* isolates of clinical and epidemiological significance were subjected to WGS. All these isolates belonged to the phylogroup Kp1, *K. pneumoniae sensu stricto*. The following 12 STs were identified: ST15 (1), ST25 (1), ST29 (2), ST39 (2), ST107 (1), ST219 (1), ST307 (2), ST395 (7), ST449 (1), ST873 (1), ST1480 (1), and ST5275 (1). This analysis indicated the circulation of international high-risk MDR clones in Armenia. In particular, 33.33% of the sequenced strains belonged to ST395, which was first detected in France in 2010 and now is emerging as the international high-risk clonal lineage of *K. pneumoniae* [25]. All four of our XDR isolates with carbapenem resistance and three ESBL-producing MDR isolates with resistance to 8 classes of AMs belonged to this ST. It is also concerning that all our ST395 isolates were recovered from a vulnerable cohort, that is, paediatric patients. This ST is associated with clinically important MDR phenotypes such as the production of carbapenemases and ESBL as well as resistance to other classes of AMs [3,4].

We also detected the representatives of other international high-risk MDR clones, ST15 [60,61] and ST307 [62]. The carbapenem-resistant MDR isolate KpA699 belonged to ST15, and two isolates with resistance to eight AM classes (KpA204 and KpA500) were assigned to ST307. Taken together with the ST395 data, our results indicate that 10 out of 18 XDR and MDR isolates belong to the recognised international high-risk MDR clones. Among other MDR isolates with resistance to nine classes of AMs, two were assigned to ST39, which is also considered an emerging pathogen [63], and one to ST219. In addition, two MDR strains, KpA250 and KpA314, belonged to ST29. Need to mention here that strains for sequencing were selected based on ERIC-PCR data, and only one clone from the clusters with identical fingerprints was selected for WGS. Thus, the prevalence of international high-risk MDR clones in our collection may be higher.

Because of our focus on hard-to-treat XDR and MDR infections, the WGS coverage of the non-MDR *K. pneumoniae* strains, which accounted for 35.42% of all isolates, remained insufficient. Nevertheless, two non-MDR HMV isolates were assigned to the STs with the hvKp phenotype. The ESBL-producer KpA828 belonged to ST25 [64], and KpA704 to ST107, while the isolate that was susceptible to AMs was assigned to ST1480. Thus, all our isolates are the well-known *K. pneumoniae* lineages disseminated in many countries. In the only previous report from the region, genomic characterisation of eight clinical isolates of *K. pneumoniae*, which were isolated in 2019, was published [24] (ENA Project: PRJEB51925). The presence of MDR ST307 isolates (N = 4) was detected as well, but also other STs, which were not present in our analysis: ST37, ST147, ST807, and ST967. These findings indicated the presence of another international high-risk MDR clone of *K. pneumoniae*, ST147 [65], in the region.

The capsule polysaccharide of *K. pneumoniae* is one of the important virulence factors, which protects against phagocytosis and the bactericidal activity of the serum [66]. There are indications that K1 and K2 serotypes may be involved in more clinically serious cases of bacteremia [67]. The lipopolysaccharide (O antigen) also plays a role as a virulence factor [68]. The most prevalent ST395 isolates shared an identical profile of capsular and LPS serotypes, K2:O2 (subtype O2a). The second most common profile, K19:O1, was found in the ST15 and ST29 isolates (14.29%, 3/21). An identical serotype, K62:O1, was also detected in two ST39 isolates. In all other isolates, the serotypes were correlated with the corresponding STs.

Further, based on WGS data, we analysed the genetic background for the phenotypically observed AMR profiles of our *K. pneumoniae* isolates. The resistome of four XDR ST395 isolates was highly similar. In particular, resistance to carbapenems was associated with the production of NDM-1 carbapenemase, while resistance to amikacin was associated with the presence of the *armA* gene. The ESBL-producer phenotype was not detected in the XDR isolates, but the combination of the *bla*_CTX-M-15_, *bla*_OXA-1_, *bla*_SHV-11,_ and *bla*_TEM-1_ genes was present, except for one isolate lacking the *bla*_CTX-M-15_ gene. The above gene combination was also detected in all MDR ST395 isolates with the ESBL-producer phenotype. There was a good concordance between the phenotypic and genotypic data: *aac*(*3*)-*IIe*, *aac*(*6*′)-*Ib*-*cr6*, and *aph*(*3*′)-VIa (aminoglycoside resistance); *mphA*, *mphE*, *mrx*, and *msrE* (macrolide resistance); *aac*(*6*′)-*Ib*-*cr6*, *gyrA* (S801), *parC* (S83I), and *qnrS1* (fluoroquinolone resistance); *dfrA1*, *dfrA5*, *sul1*, and *sul2* (resistance to folate pathway antagonists); *tet*(A)/*tetR*(A) (tetracycline resistance); *catA1* and *catB3* (phenicol resistance); *fosA6* and *uhpT* (E350Q) (fosfomycin resistance). Also, the truncated form of OmpK35 porin (72% of full length) and the lack of the *ramAR* genes, which encode the regulators of the AcrAB pump, were the characteristic features of all our XDR and MDR *K. pneumoniae* isolates belonging to ST395.

Among the other MDR isolates, the ESBL-producing phenotype was associated with the combination of genes encoding CTX-M and SHV beta-lactamases, while the production of SHV-11 was detected only in a non-MDR isolate with the HMV phenotype. Regarding resistance determinants to other classes of AMs, they were mostly identified as specific AMR gene combinations and were mostly in concordance with the phenotypic data. It must be emphasised, however, that no acquired colistin or tigecycline resistance genes can be identified in three isolates with intermediate resistance to these AMs. Although tigecycline resistance could be the result of mutations in the *tet*(A) gene [69], our three tigecycline non-susceptible isolates (KpA13, KpA204, and KpA7001) carried the same *tet*(A) gene variant as the susceptible isolates, thus ruling out this probability. Potential mechanisms could be the overexpression of the *acrB* gene and its regulator RamA due to mutations in their regulators [70] and reduced permeability of AMs due to mutations in the *ompK* genes.

The genes encoding OmpK35, OmpK36, and OmpK37 porins were present in all genomic sequences. However, the combined mutations in the *ompK36* and *ompK37* genes were identified in five (23.81%) MDR strains, and/or a truncated form of OmpK35 porin was predicted in nine (42.86%) MDR strains, suggesting that the reduced permeability to AMs may contribute to the increased MICs towards clinically important AMs in 57.14% (12/21) of *K. pneumoniae* clinical isolates. The main MDR efflux systems in *Klebsiella* spp., AcrAB and OqxAB, as well as other efflux systems (AcrEF, EefAB, EmrAB, KpnGH, KpnEF, LptD, MacAB, and MsbA) were present with a high prevalence in our clinical isolates, including the non-MDR strains. In addition, the five strains with the combined mutations in *ompK36* and *ompK37* also demonstrated an identical profile of multiple substitutions in the *acrR* gene (repressor of the AcrAB pump). These highly similar profiles of mutations in *ompK36/ompK37* and *acrR* were detected in clinical isolates belonging to different STs: ST15 (1), ST39 (2), ST395 (1), and ST5275 (1). In these isolates, the reduced permeability to AMs and overexpression of the AcrAB pump may contribute to a higher level of resistance to multiple AMs, including carbapenems.

For example, the carbapenem-resistant isolate KpA699 (ST15) harboured no known carbapenemase genes but had mutations in the *ompK36* (OmpK36GD, A217S) and *ompK37* (I70M, I128M) genes associated with resistance to carbapenems. Additionally, the strain carried a truncated OmpK35 porin, comprising 49% of the full length. In addition, the co-occurrence of mutations in the local repressor genes of the AcrAB efflux pump, *acrR* and *ramR*, was observed in this isolate. Thus, the reduced permeability of porins and overexpression of the AcrAB efflux pump may be the mechanisms of carbapenem resistance in KpA699. These non-specific resistance mechanisms may also be responsible for its resistance to cefoxitin and tetracycline, for which no acquired genetic determinants of resistance can be found.

However, the similar mutations contributing to the reduced permeability of porins and overexpression of the AcrAB efflux pump can also be found in the other *K. pneumoniae* strains, in which the specific AMR genes can be detected. Thus, these non-specific mechanisms of AMR are widespread among clinical *K. pneumoniae* strains and may potentially contribute to the elevated MIC values or interfere with therapies even if the appropriate AMs are used.

The virulome analysis revealed a high virulence potential in five out of seven ST395 isolates, including all XDR and one MDR (KpA7001) strain. These five isolates carried *ybt* locus, which encodes the siderophore yersiniabactin (lineage 16) and which is located within ICE*Kp12*; the *iucABCD* and *iutA* genes, which encode the aerobactin siderophore; and the *rmpA2* gene, which encodes the regulator of the mucoid phenotype. The additional virulence factors, *rmpADC* (mucoid phenotype) and *peg344* (metabolic transporter), were detected in one of the XDR isolates, KpA481. Notably, all these isolates had the highest score of 4 by Kleborate [39]. However, an identical truncated form of RmpA2 protein (47%) was predicted in all ST395 isolates, and they displayed the HMV-negative phenotype. This suggests that our *K. pneumoniae* ST395 isolates with the score of 4 cannot be classified as hypervirulent, based on in silico prediction of virulence biomarkers [14,15]. Still, these carbapenem-resistant or ESBL-producing strains display a high virulence potential. These highly virulent strains are not detectable by conventional laboratory tests and have to be monitored using genomic approaches. To our knowledge, this is the first report and comprehensive characterisation of carbapenem-resistant XDR and ESBL-producing MDR *K. pneumoniae* ST395 clinical isolates from Armenia carrying important virulence determinants.

Regarding the virulence potential of MDR and non-MDR *K. pneumoniae* clinical isolates, the majority of them had the *ybt* locus encoding yersiniabactin located on integrative conjugative elements (ICE*Kp*) and were scored 1 by Keborate. In addition, the virulence score of 0 was assigned to 14.29% (3/21) of the isolates. Notably, two non-MDR isolates with the HMV phenotype were scored 0 and 1 by Keborate and thus cannot be classified as hypervirulent. Given the low number of non-MDR isolates subjected to WGS, further research is needed to evaluate the prevalence of hypervirulent strains among the non-MDR *K. pneumoniae* circulating in the region.

Mobile genetic elements (MGE) play an important role in the evolution of *K. pneumoniae* lineages towards MDR and hypervirulence. The virulence loci, such as *ybt*, for example, were located within the integrative conjugative elements ICE*Kp4*, ICE*Kp9*, ICE*Kp3*, ICE*Kp5*, and ICE*Kp12* in different isolates of *K. pneumoniae.* As mentioned before, in the five ST395 isolates, several virulence determinants were located within ICE*Kp12*.

Regarding another class of MGEs, plasmids: except for one strain, KpA828, plasmid replicons were detected in all other isolates, ranging from one to seven per isolate. The highest number of replicons was identified among carbapenem-resistant and highly virulent isolates belonging to ST395, which had an identical profile of seven replicons, belonging to the next incompatibility groups: Col(pHAD28), ColRNAI, IncFIB(K), IncFIB(pNDM-Mar), IncFII(K), IncHI1B(pNDM-MAR), and IncR. Importantly, some plasmids harbour a large array of AMR genes, which, if transferred, may instantly confer resistance to a multitude of AMs. In the KpA511 strain, for example, the IncFIB(K) plasmid pCAV1099-114 carried a 6588 bp resistance region encompassing the following 10 AMR genes: *qnrS1*, *aph*(*6*)-*Id*, *aph* (*3*″)-*Ib*, *sul2*, *dfrA12*, *aadA2*, *qacEdelta1*, *sul1*, *mrx*, and *mphA*.

Prophage sequences were detected in all the sequenced isolates. All isolates belonging to ST395 did not contain any CRISPR-Cas sequences and were characterised not only by a high number of plasmid replicons but also by a high number of prophages. All these isolates, except one (KpA769), shared an identical profile of the following 6 prophages: Edward_GF_2, Escher_HK639, Klebsi_ST147_VIM1phi7.1, Klebsi_ST512_KPC3phi13.2, Klebsi_3LV2017, and Salmon_SEN34. Among isolates in our collection, all of these prophages, except for one (Escher_HK639), were detected only in ST395 isolates, suggesting the clonal dissemination of these prophages with the hosts, without horizontal transfer events to the representatives of other STs.

In nine prophage sequences, we detected the AMR genes, including the genes of resistance to the first-line drugs such as beta-lactams (6 genes). One ST395 isolate, KpA769, contained the prophage with two of these genes, with a total of eight AM genes. This suggests that the prophages in *K. pneumoniae* may serve as significant vehicles for the horizontal dissemination of AMR genes. Interestingly, the same prophages may carry a different AMR gene load in the hosts belonging to different STs. The Escher_RCS47 (NC_042128) prophage, for example, carried *qacEdelta1*, *dfrA7*, *aph*(*3*′)-*Ia*, and *catA1* in KpA13 (ST39) but *dfrA14* and *bla_CTX_*_-*M*-*15*_—in KpA250 (ST29). This again suggests the scenario of the clonal dissemination of prophage sequences with their hosts belonging to different STs. And finally, although a large-scale database analysis may suggest the presence of virulence genes in *K. pneumoniae* prophages [54], we did not detect any virulence genes in the prophage sequences of our strains. Other MGEs, such as ICEs, may be largely responsible for the horizontal dissemination of virulence genes.

Thus, our results revealed the circulation of international high-risk MDR clones of *K. pneumoniae* in Armenia. They are associated with an elevated risk of treatment failure due to difficult-to-treat AMR phenotypes and pose a considerable threat to patients, especially in vulnerable cohorts such as children. Particularly problematic are the clonally related NDM-1 carbapenemase-producing XDR and ESBL-producing MDR *K. pneumoniae* ST395 strains. They display a high virulence potential, harbour multiple plasmid replicons and prophages, and are associated with an increased risk of nosocomial infections. Especially concerning is the finding of carbapenem-resistant XDR strains in infected children. While some non-carbapenem antimicrobials, such as colistin, are utilised for treating carbapenem-resistant infections, their use is typically restricted to adult patients due to the side effects. Another restriction is that colistin and tigecycline are not licensed for use in Armenia. Regrettably, resistance to these antibiotics is on the rise, further complicating treatment options for carbapenem-resistant infections [70,71]. We also found resistance to these antibiotics in our three *K. pneumoniae* isolates.

Despite these challenges, recent research efforts have resulted in the development of several new drugs and combination therapies, which demonstrated good efficacy against carbapenem-resistant bacteria [21]. Some of these promising agents include ceftazidime-avibactam, cefiderocol, ceftolozane-tazobactam, imipenem-cilastatin-relebactam, meropenem-vaborbactam, plazomicin, and eravacycline. Unfortunately, these drugs are currently not available for clinical use in Armenia.

The prevalence of international high-risk MDR clones among the clinical isolates of *K. pneumoniae* in Armenia requires further monitoring and control measures. One of the limitations of the current analysis, however, is due to the lack of sufficient genomic data from the neighbouring countries, with which active interaction in the form of travel is maintained, such as Georgia and Iran. Nevertheless, our findings underscore the pressing need for genomic surveillance of *K. pneumoniae* infections of epidemiological significance in the country and beyond. Such surveillance efforts are crucial for improving AM therapy strategies and for the identification of intervention points to limit the dissemination of MDR *K. pneumoniae*.

## 4. Materials and Methods

### 4.1. Human Isolates of K. pneumoniae

This study was performed with the use of a collection of *K. pneumoniae* strains isolated from patients in three hospitals in Armenia between 2018 and 2024. A total of 48 non-duplicate *K. pneumoniae* isolates were isolated from the stool (17), urine (16), throat (10), endotracheal tube (3), eye (1), and wound fluid (1) samples. The majority of these isolates, 38 (79.17%), were obtained from paediatric patients, and the rest 10 (20.83%)—from adults. Clinical and microbiological data of these patients were collected and anonymised. The gender distribution was as follows: 25 males (52.08%) and 23 females (47.92%). A total of 37 (77.08%) isolates were collected from patients who had not taken any medications, including antibiotics, before the hospital admission. In addition, 11 (22.92%) strains were isolated from patients receiving treatment with third-generation cephalosporins. Among these, three isolates were from the endotracheal tubes in paediatric patients and eight were isolates from the urine of adult patients. Of the 48 patients in this study, 31 (64.58%) and 17 (35.42%) were from Yerevan and regions, respectively. Additional information regarding patients and *K. pneumoniae* isolates is presented in Appendix A.

Ethical Statement. The study protocol was approved by the Ethics Committee of the Institute of Molecular Biology NAS RA (IORG number 0003427, IRB/IEC: 00004079); protocol code: Approval 01/2017, date of approval: 14 June 2017; protocol code: Approval 05/23, date of approval: 31 October 2023.

### 4.2. Antimicrobial Susceptibility Testing

All 48 *K. pneumoniae* isolates were tested for susceptibility towards 22 individual antibiotics, which covered 12 different classes of antibiotics. The SOPs were strictly followed, in accordance with the guidelines of the Clinical and Laboratory Standards Institute (CLSI) for standard disc diffusion assays [72]. For this assay, Muller–Hinton agar (Liofilchem^®^ s.r.l., Roseto degli Abruzzi, Italy) was used. Bacterial inoculum was adjusted to the equivalent of a 0.5 McFarland standard. The following AM discs (Liofilchem^®^ s.r.l., Roseto degli Abruzzi, Italy) were used: amikacin (30 µg), amoxicillin-clavulanic acid (20 µg/10 µg), ampicillin (10 µg); ampicillin-sulbactam (10 µg/10 µg), aztreonam (30 µg), cefepime (30 µg), cefoxitin (30 µg), ceftazidime (30 µg), ceftriaxone (30 µg), chloramphenicol (30 µg), ciprofloxacin (5 µg), gentamicin (10 µg), imipenem (10 µg), meropenem (10 µg), piperacillin-tazobactam (100 µg/10 µg), tetracycline (30 µg), ticarcillin-clavulanate (75 µg/10 µg), tobramycin (10 µg), and trimethoprim-sulfamethoxazole (1.25 µg/23.75 µg). The results of susceptibility testing were interpreted based on the CLSI criteria [72]. The MIC for azithromycin was determined by performing the agar dilution method according to the CLSI standards [73]. For determination of MIC for colistin, the broth disc elution method was used, according to CLSI recommendations [73]. The MIC for tigecycline (Sigma–Aldrich, St. Louis, MO, USA) was determined by using the broth dilution method. The MIC values were assessed in accordance with the European Committee on Antimicrobial Susceptibility Testing (EUCAST) criteria: ≤0.5 mg/L was considered susceptible and >0.5 mg/L—resistant [74]. The ESBL-producer phenotype was identified by the disc diffusion test using cefotaxime and ceftazidime with and without clavulanic acid, according to the CLSI guidelines [72]. *Escherichia coli* strains ATCC 25922 and ATCC 35218 were used for quality control. Isolates that showed resistance to representatives of at least three classes of AMs were considered as MDR, and isolates non-susceptible to ≥1 AM agent in all but ≤2 classes of AMs were considered as XDR [75].

### 4.3. Susceptibility to Bacteriophage Preparations

Bacteriophage susceptibility of *K. pneumoniae* clinical isolates was assessed with the “streak assay” [76]. The commercial bacteriophage preparations “Bacteriophage *Klebsiella pneumoniae* Purified” (BKpP) and “Bacteriophage *Klebsiella* Polyvalent Purified” (BKPP) from (SPA “Microgen” Moscow, Russia) were used.

### 4.4. Hypermucoviscous (HMV) Phenotype Identification

The “string test” to determine the HMV phenotype of *K. pneumoniae* isolates was performed as described previously [12]. The isolates were inoculated on agar plates and incubated at 37 °C overnight. HMV isolates were identified by the formation of a viscous string from the colonies on the agar surface to an inoculating loop measuring at least 5 mm.

### 4.5. Bacterial DNA Extraction

Total bacterial DNA samples for ERIC-PCR analysis were isolated by the boiling lysate method [77] and frozen at −20 °C until further analysis. For whole genome sequencing (WGS), bacterial DNA samples were extracted using the UltraClean^®^ Microbial DNA Isolation Kit (MO BIO Laboratories Inc., San Diego, CA, USA) according to the manufacturer’s recommendations. DNA samples were stored in the 10 mM Tris buffer, without EDTA, at −20 °C.

### 4.6. ERIC-PCR

The primers used for ERIC-PCR were: ERIC-1R (5′-ATGTAAGCTCCTGGGGATTCAC-3′) and ERIC-2 (5′-AAGTAAGTGACTGGGGTGAGCG-3′) (Integrated DNA Technologies, BVBA—Löwen, Belgium) [78]. PCR was performed as described previously [79], with some modifications. The PCR conditions were as follows: initial denaturation at 94 °C for 4 min, followed by 35 cycles of denaturation at 94 °C for 1 min, primer annealing at 52 °C for 1 min, an extension at 72 °C for 4 min, and a final extension at 74 °C for 10 min. The amplified products were separated by gel electrophoresis in 1.5% agarose. HyperLadder™ 1 kb (Bioline, Memphis, TN, USA) was used as a molecular weight marker.

The amplicon patterns generated by ERIC-PCR were analysed with the gel analysis software GelAnalyzer 19.1 (www.gelanalyzer.com). After normalisation and pattern alignment, the dendrogram showing the amplicon similarity among isolates was generated with the Dice coefficient and the unweighted pair group method with arithmetic average (UPGMA) algorithm for cluster analysis (http://insilico.ehu.eus/dice_upgma, accessed on 20 September 2024).

### 4.7. WGS of K. pneumoniae Isolates

WGS of 21 *K. pneumoniae* isolates in this study was provided by MicrobesNG (https://microbesng.com). Sequencing of two isolates from 2018 and 2019 (KpA500 and KpA6101) was performed on the Illumina HiSeq 2500, and the other 19 isolates were sequenced on the Illumina NovaSeq 6000 platform. Sequencing was performed with 2 × 250 bp paired-end reads at 30× coverage. Reads were adapter-trimmed using Trimmomatic 0.30 [80] with a sliding window quality cutoff of Q15. Contigs were annotated using Prokka 1.11 [81]. Whole genome sequences of *K. pneumoniae* isolates are available in the NCBI database under Bioproject/PRJNA1141898. Accession numbers for individual isolates are listed in Appendix A.

### 4.8. Bioinformatics Analyses

The general information on the genomes of our *K. pneumoniae* isolates was obtained using the Pathogenwatch resource (https://pathogen.watch, accessed on 20 September 2024) and BIGSdb-Pasteur database (https://bigsdb.pasteur.fr/klebsiella/, accessed on 20 September 2024). Assignment of the isolates to STs and cgMLST was performed using the BIGSdb-Pasteur database (https://bigsdb.pasteur.fr/klebsiella/, accessed on 20 September 2024). Capsule (K) type and O serotype of the *K. pneumoniae* isolates were identified using the Kaptive tool [26] (https://kaptive-web.erc.monash.edu, accessed on 20 September 2024). Virulence scores by Kleborate [39] were obtained using the Pathogenwatch resource (https://pathogen.watch, accessed on 20 September 2024). Prediction of known or potential virulence factors in silico was performed using the Virulence Factor Database (VFDB) [49] (http://www.mgc.ac.cn/cgi-bin/VFs/v5/main.cgi, accessed on 20 September 2024). Antibiotic resistance genes were identified using the Resistance Gene Identifier (RGI) tool in the comprehensive antibiotic resistance database (CARD) [52] (https://card.mcmaster.ca/analyze/rgi, accessed on 20 September 2024), the ResFinder v.4.6.0 tool [35,36] (http://genepi.food.dtu.dk/resfinder, accessed on 20 September 2024), and the BIGSdb-Pasteur database (https://bigsdb.pasteur.fr/klebsiella/, accessed on 20 September 2024). Detection of plasmid replicons and determination of incompatibility groups was performed using the PlasmidFinder 2.1 tool [36,42] (https://cge.food.dtu.dk/services/PlasmidFinder/, accessed on 20 September 2024). Identification of mobile genetic elements and their linkage with AMR genes was performed with MobileElementFinder (v1.0.2) [82] (https://cge.food.dtu.dk/services/MobileElementFinder, accessed on 20 September 2024). Identification and annotation of prophage sequences within bacterial genomes was performed using the PHASTEST web server [43,44,45] (https://phastest.ca/, accessed on 20 September 2024). CRISPR sequences were identified using the Pathosystems resource integration centre (PATRIC) [46] (https://www.patricbrc.org, accessed on 20 September 2024). Other genes in contigs were analysed using the BLAST server (http://blast.ncbi.nlm.nih.gov/Blast.cgi, accessed on 20 September 2024). The Average Nucleotide Identity (ANI) value was determined using the ANI Calculator tool (https://www.ezbiocloud.net/tools/ani, accessed on 20 September 2024).

### 4.9. Whole Genome-Based Phylogenetic Analyses

Whole-genome-based phylogenetic trees of *K. pneumoniae* isolates were obtained using the type strain genome server (TYGS) platform (https://tygs.dsmz.de, accessed on 20 September 2024) [55,56]. Pairwise comparison of genomes was performed using the genome blast distance phylogeny (GBDP) method, and intergenomic distances were inferred as described earlier [55]. Phylogenetic trees were constructed with FastME 2.1.6.1 [56].

The Pathogenwatch resource (https://pathogen.watch, accessed on 20 September 2024) was used to generate phylogenetic trees of *K. pneumoniae* strains based on core genome distances. The dendrograms were constructed based on scaled pairwise scores for assemblies and the neighbour-joining method (APE package [83]). The phangorn package [84] was used to obtain the midpoint rooted tree.

Tree annotation and visualisation were performed using iTol v.6 [85] (https://itol.embl.de/, accessed on 20 September 2024).

### 4.10. Statistical Analyses

The *p*-value (two-tailed) from Fisher’s exact test was calculated using the online GraphPad QuickCalcs resource (http://www.graphpad.com/quickcalcs/contingency1.cfm, accessed on 20 September 2024) to evaluate statistical differences between the compared groups. *p*-values *≤* 0.05 were considered significant.

## 5. Conclusions

We performed a comprehensive analysis of *K. pneumoniae* pathotypes in Armenia, and the main conclusions are:The majority (64.58%) of clinical *K. pneumoniae* isolates are represented by the XDR and MDR strains, with resistance from five to ten AM classes. Only 35.42% of the isolates are resistant to less than five AM classes.Phage therapy could be a viable option for an alternative/adjunct therapy of XDR and MDR isolates of *K. pneumoniae*.Epidemiologically, the most problematic *K. pneumoniae* lineages are represented by international high-risk MDR clones belonging to ST395, ST15, and ST307.The XDR and MDR strains demonstrate a high virulence potential, with a number of virulence determinants ranging from capsule polysaccharides to siderophores to regulators of the mucoid phenotype.In part, AMR mechanisms in *K. pneumoniae* are non-specific and driven by mutations in the porin genes, which reduce permeability to AMs, and by mutations in the regulators of efflux pumps, which allow overexpression of drug efflux pumps such as AcrAB. These mechanisms are responsible for AMR in strains with the apparent absence of specific AMR genes.*K. pneumoniae* isolates possess an extensive range of MGEs, ranging from ICEs to plasmids to prophages, especially in ST395 strains.Many AMR and virulence genes are located on MGEs, which may allow rapid evolution towards MDR and hypervirulent traits in these bacteria.The overall situation with *K. pneumoniae* pathotypes in Armenia dictates the urgent need for genomic surveillance of this infection, especially in light of the emergence of global hypervirulent STs such as hvKp ST23 (https://www.who.int/emergencies/disease-outbreak-news/item/2024-DON527, accessed on 20 September 2024).

## Figures and Tables

**Figure 1 ijms-26-00504-f001:**
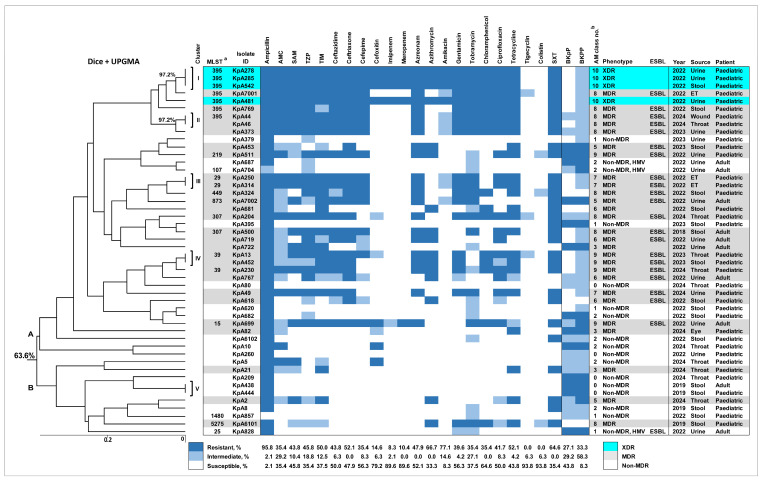
Dendrogram of ERIC-PCR fingerprinting profiles of 48 *K. pneumoniae* clinical isolates, with sequence types (MLST) and susceptibility to AMs and phage cocktails. The dendrogram was generated with Dice coefficient and the UPGMA clustering method; A and B — main clades; I, II, III, IV and V—cluster numbers. ^a^—identified based on WGS data, ^b^—number of classes of AMs to which isolate displayed resistance. Abbreviations: AMC, amoxicillin-clavulanate; TIM, ticarcillin-clavulanate; TZP, piperacillin-tazobactam; SAM, ampicillin-sulbactam; SXT, trimethoprim-sulfamethoxazole; BKpP, “Bacteriophage *Klebsiella pneumoniae* Purified”; BKPP, “Bacteriophage *Klebsiella* Polyvalent Purified”; AM, antimicrobial; ESBL, extended spectrum beta-lactamase; XDR, extensively drug-resistant; MDR, multidrug resistant; HMV, hypermucoviscous; ET, endotracheal tube.

**Figure 2 ijms-26-00504-f002:**
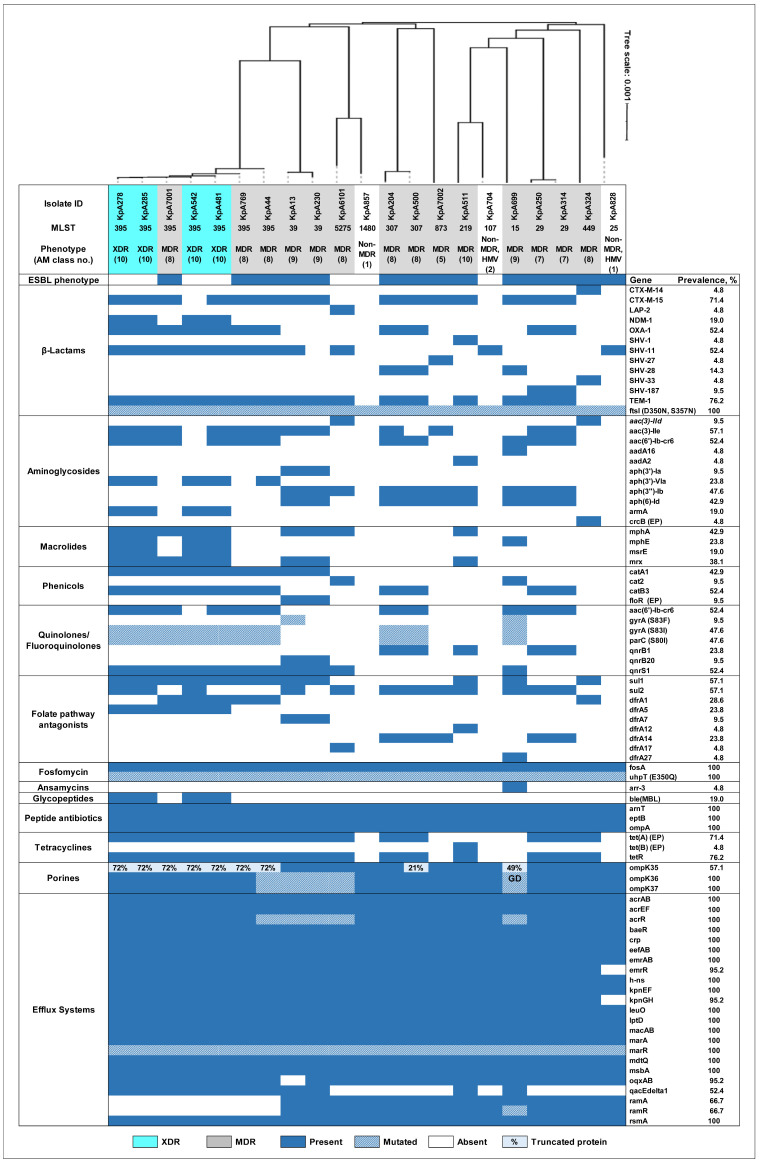
WGS-based phylogenetic tree of 21 *K. pneumoniae* clinical isolates, with MLST, AMR phenotypes, and AMR-associated genes. Tree was obtained using type strain genome server (TYGS) (https://tygs.dsmz.de, accessed on 20 September 2024). Abbreviations: AM, antimicrobial; EP, efflux pump; ESBL, extended spectrum beta-lactamase; GD, Gly134Asp135 duplication in loop 3 of OmpK36; HMV, hypermucoviscous; MDR, multidrug resistant; MLST, multilocus sequence type; XDR, extensively drug-resistant.

**Figure 3 ijms-26-00504-f003:**
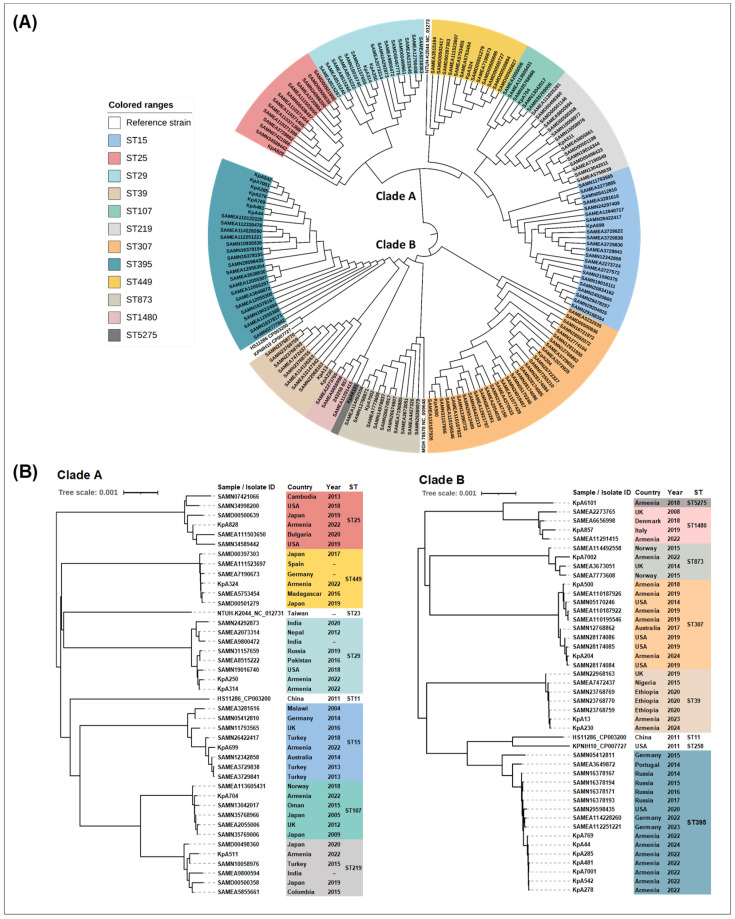
WGS-based phylogenetic tree of *K. pneumoniae* clinical isolates. (**A**) Phylogenetic tree inferred from core genome analysis using Pathogenwatch resource (https://pathogen.watch, accessed on 20 September 2024); (**B**) WGS-based tree of clade A and clade B strains obtained using TYGS platform [56,57] (https://tygs.dsmz.de, accessed on 20 September 2024). Tree annotation and visualisation were performed using iTol v.6 (https://itol.embl.de/, accessed on 20 September 2024). “─” no available information. Abbreviation: ST, multilocus sequence type.

**Table 1 ijms-26-00504-t001:** Molecular epidemiology of 21 *K. pneumoniae* clinical isolates.

Isolate ID	MLST ^1^	cgMLST ^1^	Sub- Lineage ^1^	Clonal Group ^1^	K Locus/ K Serotype ^2^	O Locus; O Serotype ^2^
KpA699	15	*c9eb	15	15	KL19/K19	O1/O2v2; O1
KpA828	25	*727e	25	25	KL2/K2	Unknown (O3/O3a); Unknown (O3/O3a)
KpA250	29	*a192	29	10,208	KL19/K19	O1/O2v2; O1
KpA314	29	*a192	29	10,208	KL19/K19	O1/O2v2; O1
KpA13	39	*3bbc	39	10,192	KL62/K62	O1/O2v2; O1
KpA230	39	*ffef	39	10,192	KL62/K62	O1/O2v2; O1
KpA704	107	*de77	107	10,012	KL10/K10	O1/O2v1; O1
KpA511	219	*74a7	107	219	KL114/Unknown (not serologically defined)	O1/O2v1; O1
KpA204	307	1935	307	307	KL102/Unknown (not serologically defined)	O1/O2v2; o2afg
KpA500	307	*fb39	307	307	KL102/Unknown (not serologically defined)	O1/O2v1/O2a
KpA44	395	*72e4	395	395	KL2/K2	O1/O2v2; O1
KpA278	395	*a23d	395	395	KL2/K2	O1/O2v1; O2a
KpA285	395	*a23d	395	395	KL2/K2	O1/O2v1; O2a
KpA481	395	*72e4	395	395	KL2/K2	O1/O2v1; O2a
KpA542	395	*a23d	395	395	KL2/K2	O1/O2v1; O2a
KpA7001	395	5169	395	395	KL2/K2	O1/O2v1; O2a
KpA769	395	*70fc	395	395	KL2/K2	O1/O2v1; O2a
KpA324	449	*2c01	35	35	KL22/K22	O1/O2v1; O1
KpA7002	873	*9b78	873	873	KL52/K52	OL101; unknown (OL101)
KpA857	1480	7702	37	1480	KL39/K39	O3b; O3b
KpA6101	5275	6826	37	10,094	KL23/K23	O1/O2v2; O2afg

^1^ Sourced from the BIGSdb-Pasteur database (https://bigsdb.pasteur.fr/klebsiella/, accessed on 20 September 2024); ^2^ Sourced from the Kaptive [26] (https://kaptive-web.erc.monash.edu, accessed on 20 September 2024). * Novel core genome sequence type. Abbreviations: MLST, multilocus sequence type; cgMLST, core genome multilocus sequence type.

**Table 2 ijms-26-00504-t002:** Virulence biomarkers, plasmid replicons, prophages, and CRISPR arrays predicted for 21 *Klebsiella pneumoniae* clinical isolates.

Isolate ID	MLST ^1^	Virulence [39]		Plasmid Replicons ^2^ (N=)		Prophages ^3^ N=	CRISPR Arrays ^4^ N=
		Biomarkers	Score				
KpA699	15	*ybt* 16 (ICE*Kp*12)	1	Col(pHAD28); ColpVC; IncFIA(HI1); IncR	(4)	5	2
KpA828	25	*ybt* 9 (ICE*Kp*3)	1	-	(0)	2	-
KpA250	29	*ybt* 14 (ICE*Kp*5)	1	Col440I; IncFIA (pBK30683); IncFIB(K); IncFII(K)	(4)	3	-
KpA314	29	*ybt* 14 (ICE*Kp*5)	1	Col440I; IncFIA (pBK30683); IncFIB(K); IncFII(K)	(4)	3	-
KpA13	39	*ybt* 15 (ICE*Kp*11)	1	Col440I; IncFIB(K) (pCAV1099-114); IncR	(3)	3	1
KpA230	39	*ybt* 15 (ICE*Kp*11)	1	Col440I; IncFIB(K) (pCAV1099-114); IncR	(3)	1	1
KpA704	107	-	0	IncFIB(K); IncFII(K)	(2)	4	-
KpA511	219	*ybt* 14 (ICE*Kp*5)	1	IncFIB(K) (pCAV1099-114); IncR	(2)	5	-
KpA204	307	-	0	IncFIB(K)	(1)	4	-
KpA500	307	*ybt* 1 (ICE*Kp*4)	1	IncFIB(K)	(1)	3	-
KpA44	395	*ybt* 16 (ICE*Kp*12)	1	Col (pHAD28); ColRNAI; IncQ1; IncR	(4)	6	-
KpA278	395	*ybt* 16 (ICE*Kp*12); *iuc* 1; *rmpA2*_6*-47%	4	Col (pHAD28); ColRNAI; IncFIB(K); IncFIB (pNDM-Mar); IncFII(K); IncHI1B (pNDM-MAR); IncR	(7)	6	-
KpA285	395	*ybt* 16 (ICE*Kp*12); *iuc* 1; *rmpA2*_6*-47%	4	Col (pHAD28); ColRNAI; IncFIB(K); IncFIB (pNDM-Mar); IncFII(K); IncHI1B (pNDM-MAR); IncR	(7)	6	-
KpA481	395	*ybt* 16 (ICE*Kp*12); *iuc 1*; *rmp1*; KpVP-1/*rmpA2*_6*-47%	4	Col (pHAD28); ColRNAI; IncFIB (pNDM-Mar); IncR	(4)	6	-
KpA542	395	*ybt* 16 (ICE*Kp*12); *iuc* 1; *rmpA2*_6*-47%	4	Col (pHAD28); ColRNAI; IncFIB(K); IncFIB (pNDM-Mar); IncFII(K); IncHI1B (pNDM-MAR); IncR	(7)	6	-
KpA769	395	*ybt* 16 (ICE*Kp*12)	4	Col (pHAD28); ColRNAI; IncFIB(K); IncFII(K); IncR	(5)	6	-
KpA7001	395	*ybt* 16 (ICE*Kp*12); *iuc* 1; *rmpA2*_6*-47%	1	Col (pHAD28); ColRNAI; IncFIB(K); IncFIB (pNDM-Mar); IncFII(K); IncHI1B(pNDM-MAR); IncR	(7)	5	-
KpA324	449	*ybt* 1 (ICE*Kp*4)	1	Col (pHAD28); IncFIB(K)	(2)	2	2
KpA7002	873	*ybt* 8 (ICE*Kp*9)	1	IncFIB(K)	(1)	5	1
KpA857	1480	*ybt* 4 (plasmid)	1	IncFIB(K) (pCAV1099-114)	(1)	4	-
KpA6101	5275	-	0	IncR	(1)	2	-

^1^ Sourced from the BIGSdb-Pasteur database (https://bigsdb.pasteur.fr/klebsiella/, accessed on 20 September 2024); ^2^ PlasmidFinder 2.1 tool [36,42] (https://cge.food.dtu.dk/services/PlasmidFinder/, accessed on 20 September 2024); ^3^ PHASTEST web server [43,44,45] (https://phastest.ca/, accessed on 20 September 2024); ^4^ Pathosystems resource integration centre (PATRIC) [46] (https://www.patricbrc.org, accessed on 20 September 2024). Abbreviations: MLST, multilocus sequence type; CRISPR, clustered regularly interspaced short palindromic repeats.

**Table 3 ijms-26-00504-t003:** Prophage regions containing AMR genes in 6 *K. pneumoniae* clinical isolates.

Isolate ID	MLST	Prophage	Intact/Questionable ^1^	AMR Genes
KpA13	39	Escher_RCS47 (NC_042128)	Intact	*qacEdelta1*, *dfrA7*, *aph*(*3*′)-*Ia*, *catA1*,
		Microc_MaMV_DC (NC_029002)	Intact	*qnrS1*, *aph*(*6*)-*Id*, *aph*(*3*″)-*Ib*, *sul2*
KpA204	307	Staphy_SPbeta_like (NC_029119)	Intact	*dfrA14*, *aac*(*3*)-*IIe*, *catB3*, *bla*_OXA-1_, *aac*(*6*′)-*Ib*-*cr6*, *tet*(*A*), *tetR*, *qnrB1*
KpA250	29	Escher_RCS47 (NC_042128)	Intact	*dfrA14*, *bla*_CTX-M-15_
KpA511	219	Salmon_SJ46 (NC_031129)	Questionable	*bla* _TEM-1_
KpA699	15	Klebsi_ST15_OXA48phi14.1 (NC_049454)	Intact	*adeF*
		Klebsi_phiKO2 (NC_005857)	Intact	*sul1*, *qacEdelta1*, *aadA16*, *dfr27*, *arr*-*3*, *aac*(*6*′)-*Ib*-*cr6*
		Klebsi_phiKO2 (NC_005857)	Questionable	*bla*_TEM-1_, *sul2*, *aph*(*6*)-*Id*, *aph*(*3*″)-*Ib*
KpA769	395	Escher_RCS47 (NC_042128)	Questionable	*dfrA1*, *qacEdelta1*, *sul1*, *aac*(*6*′)-*Ib*-*cr6*, *bla*_OXA-1_, *catB3*, *aac*(*3*)-*IIe*, *bla*_CTX-M-15_

^1^ Regions were classified using criteria from PHASTEST [43,44,45]. Abbreviations: MLST, multilocus sequence type; AMR, antimicrobial resistance.

## Data Availability

Data associated with this article are included in the Appendix A. Whole genome sequences of *K. pneumoniae* isolates are available in the NCBI database under Bioproject/PRJNA1141898. Accession numbers for individual isolates are listed in the Appendix A.

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
