# Peer review of "Molecular Epidemiology and In-Depth Characterization of Klebsiella pneumoniae Clinical Isolates from Armenia"

_ijms, 2025, doi:10.3390/ijms26020504_

Round 1

Reviewer 1 Report

Comments and Suggestions for Authors

This manuscript presents a comprehensive study of the molecular epidemiology and characterization of Klebsiella pneumoniae clinical isolates from Armenia. The authors analyze 48 clinical isolates collected between 2018 and 2024, focusing on antimicrobial resistance (AMR), genomic diversity, and the presence of hypervirulent strains. This timely research addresses a significant public health concern regarding multidrug-resistant pathogens, particularly in regions where data is scarce. Below are some comments on how to improve the quality of the manuscript.

Lines 631 - 641: This paragraph appears to be an interpretation of the data. It is better to move it to the Discussion section.

Lines 840 - 845: This paragraph addresses a limitation in the study. It would be better to move it to the end of the Discussion section.

Lines 847 - 859: This paragraph seems to belong to the Introduction section.

Lines 876 - 878: Is there some clinical data supporting the authors’ suggestion to use cefoxitin as an alternative to carbapenems to treat ESBL-producer K. pneumoniae?

Line 965: “CTX-M” is misspelled here as “CXT-M.”

Line 1039: There is no antibiotic called “cefepime-tazobactam”. I believe the authors meant “Ceftolozane-tazobactam.”

Overall, the Discussion section reiterates the data mentioned in the Results section rather than providing a contextual interpretation of the data. The authors should more thoroughly address the implications of their findings for local healthcare practices and policies.

The conclusion only summarizes the findings of the study. It should include recommendations for surveillance and control measures to strengthen the practical relevance of the study.

Reviewer 2 Report

Comments and Suggestions for Authors

The manuscript analyzes the issue of antimicrobial resistance (AMR) in clinical isolates of Klebsiella pneumoniae from Armenia. The study provides comprehensive data and addresses gaps in similar research from the region, offering significant clinical value.

My suggestions are as follows:

1.Please add error bars and significance markers in Figure 1 to more clearly convey the statistical results.

2.Use different colors to indicate the distribution of MDR and XDR isolates.

3.For the association between resistance genes, virulence genes, and resistance phenotypes, it is recommended to use heatmaps or network diagrams for visualization.

4.Provide transcriptional-level qPCR or protein expression analyses to further elucidate the association between the NDM-1 gene and high resistance.

5.Supplement patient backgrounds (e.g., prior treatment history) and environmental factors that may contribute to antimicrobial resistance.

6.In the conclusion section, please summarize the clinical significance of the findings and propose key directions for future research on resistance and virulence.

Round 2

Reviewer 2 Report

Comments and Suggestions for Authors

I don't have any more comments after the author's revisions.